# MMSite: A Multi-modal Framework for the Identification of Active Sites in Proteins

**Song Ouyang**[1]    **Huiyu Cai**[2,3,4]    **Yong Luo**[1*]   **Kehua Su**[1*]   **Lefei Zhang**[1]    **Bo Du**[1]

[1]School of Computer Science, National Engineering Research Center for Multimedia Software
and Institute of Artificial Intelligence, Wuhan University, China
[2]BioGeometry, China    [3]Mila - Québec AI Institute, Canada    [4]Université de Montréal, Canada
{ouyangsong,luoyong,skh,zhanglefei,dubo}@whu.edu.cn
huiyu.cai@mila.quebec

## Abstract

The accurate identification of active sites in proteins is essential for the advancement of life sciences and pharmaceutical development, as these sites are of critical importance for enzyme activity and drug design. Recent advancements in protein language models (PLMs), trained on extensive datasets of amino acid sequences, have significantly improved our understanding of proteins. However, compared to the abundant protein sequence data, functional annotations, especially precise per-residue annotations, are scarce, which limits the performance of PLMs. On the other hand, textual descriptions of proteins, which could be annotated by human experts or a pretrained protein sequence-to-text model, provide meaningful context that could assist in the functional annotations, such as the localization of active sites. This motivates us to construct a **ProT**ein-**A**ttribute text **D**ataset (**ProTAD**), comprising over 570,000 pairs of protein sequences and multi-attribute textual descriptions. Based on this dataset, we propose **MMSite**, a multi-modal framework that improves the performance of PLMs to identify active sites by leveraging biomedical language models (BLMs). In particular, we incorporate manual prompting and design a MACross module to deal with the multi-attribute characteristics of textual descriptions. MMSite is a two-stage ("First Align, Then Fuse") framework: first aligns the textual modality with the sequential modality through soft-label alignment, and then identifies active sites via multi-modal fusion. Experimental results demonstrate that MMSite achieves state-of-the-art performance compared to existing protein representation learning methods. The dataset and code implementation are available at `https://github.com/Gift-OYS/MMSite`.

## 1   Introduction

The identification of active sites in proteins is crucial for advancing fields such as life sciences and pharmaceutical development. Active sites are specific regions within a protein where substrate molecules undergo chemical transformations. Understanding these sites is essential for elucidating enzyme mechanisms, designing inhibitors, and developing novel drugs. Traditionally, the recognition of these sites relied on crystallographic techniques, mass spectrometry, and other labor-intensive experiments. Recently, the advent of deep learning has marked in a new era of bioinformatics, significantly enhancing the capabilities for predicting and analyzing protein functions computationally.

Recent advancements in natural language processing (NLP), particularly with the introduction of large-scale language models (LLMs) [39] [54] [53], have revolutionized the interpretation of complex data. Inspired by these developments, protein language models [12] [2], trained on extensive datasets

---

*Corresponding authors.

38th Conference on Neural Information Processing Systems (NeurIPS 2024).

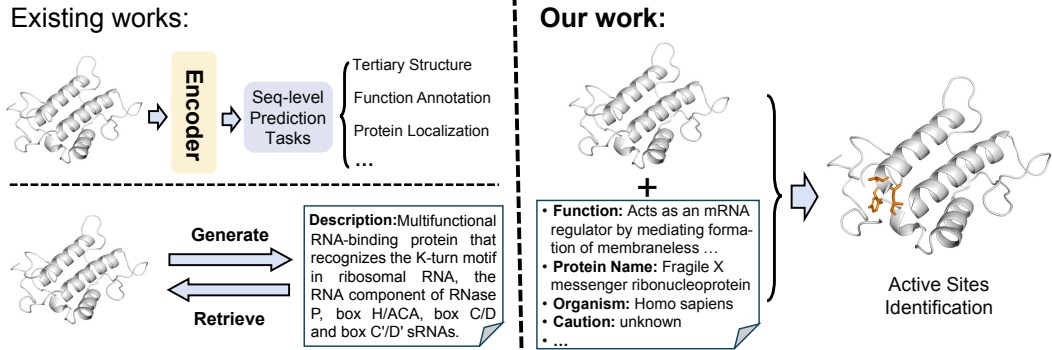

Figure 1: Difference between our work and existing mainstream works. Left: Existing works focus on obtaining comprehensive protein representations for sequence-level prediction tasks, generating texts from sequence, or retrieving sequences based on textual descriptions. Right: Our task aims to identify active sites at the residue-level using protein sequences and multi-attribute textual descriptions.

of amino acid sequences, have significantly advanced our understanding of the intricate language of proteins. Most of the existing studies focus on utilizing PLMs to capture the overall structure and function, and predicting the global properties ("fitness") of proteins [19]. In contrast, predicting properties at residue level for a given protein (such as identifying protein active sites) by deep learning methods is biologically meaningful but relatively less studied unfortunately, due to the scarcity of precise per-residue annotations.

On the other hand, informative textual descriptions, which can be obtained from biological experiments or a pretrained protein sequence-to-text model [1] [32], are widely accessible and anticipated to provide meaningful context that could assist in residue-level tasks. Moreover, the field of multi-modal deep learning [38], which combines diverse data modalities like image and text, provides promising methodologies for advancing protein research. Inspired by this, integrating protein sequences and their corresponding textual descriptions enables us to leverage the strengths of both data types to achieve a more comprehensive understanding of proteins [61] [42] (Figure 1). Our task, multi-modal protein active sites identification, is formally similar to multi-modal named entity recognition (MNER) [36] [31] which combines text and image inputs to identify named entities, but the former is inherently more challenging because: (1) MNER deals with textual and image data, whose interactions are more intuitive and well-studied, whereas the relationship between the amino acid sequence and the textual descriptions of a protein is more implicit; (2) Identifying the active sites requires a detailed understanding of both the protein sequence and its corresponding structure, which are not readily available during inference; (3) While there is a vast amount of text and image data available for MNER, high-quality datasets for multi-modal active sites identification are less abundant.

In this paper, we build a **ProT**ein-**A**ttribute text **D**ataset (**ProTAD**) containing more than 570,000 protein-text description pairs. Each textual description of a protein contains 17 different attribute fields, providing a rich semantic representation of the protein. Then, we develop a novel framework, **MMSite**, to achieve active site identification by leveraging pretrained PLMs and BLMs. Specifically, we employ prompting and design a multi-attribute cross attention module, MACross, to process the text, and achieve the identification via soft-label alignment and multi-modal fusion. Our method enforces the distribution of textual modality to be close to that of the sequential modality, improving the identification performance of PLMs. During the inference stage, we assume that there are no text descriptions, so the input is only the protein sequence, and the missing textual modality is generated with the aid of an agent model. Extensive experiments demonstrate that our method outperforms existing protein representation learning methods across three token-level and two region-level metrics, and can be seamlessly integrated with different PLMs and BLMs.

Our contributions can be summarized as follows:

- We propose a new and meaningful task in biological science: identifying active sites in proteins using both sequence and textual descriptions, and construct the ProTAD dataset.

- We introduce a framework that integrates both modalities for predicting protein active sites, utilizing a "First Align, Then Fuse" strategy.

- Our comprehensive experiments verify the effectiveness of our approach, and demonstrate that our framework can be effectively applied to different PLMs and BLMs.

## 2 Related work

**Protein representation learning** Significant progress has been made in the field of protein representation learning (PRL) due to the advancements in deep learning techniques. Graph neural networks (GNNs) have emerged as powerful tools for representing protein structures or sequences by encoding them as graphs, capturing intricate interaction patterns among proteins [17] [68]. Additionally, self-supervised learning methods have been widely adopted, utilizing various predictive tasks to train models to learn meaningful representations of proteins. The advent of protein language models like ESMs [34] [18] [30], ProteinBERT [2], TAPE [44], and ProtTrans[12], trained on vast databases of protein sequences [50] [47] [48] and structures [24] [56], has shown promising results in downstream tasks. They demonstrate the potential of transfer learning in protein representation [19]. Moreover, all-atom structures [22] [69] and protein surfaces [14] [51] have also been explored to enhance our understanding of protein structure and function. These studies have applications in protein structure prediction [24] and functional annotation [7] [5]. Despite these advances, there remains a relative scarcity of work focusing on residue-level protein understanding, which plays a crucial role in comprehending the biochemical mechanism of proteins and drug discovery.

**Multi-modal representation learning** Multi-modal representation learning addresses the challenge of effectively integrating and utilizing information from diverse data modalities. In this field, significant advancements have been achieved by developing advanced algorithms and models, such as BLIP2 [27] and PaLM-E [11], to improve the learning capabilities. Additionally, the introduction of highly capable language models, notably LLMs like GPT-4 [39], LLaMA2 [54], and Alpaca [53], has ignited fresh enthusiasm in the simultaneous modeling of biomolecules and natural language [41] concurrently. In the field of proteins, while "sequence-structure" multi-modal learning is successful [67], incorporating texts is also gain popularity. For example, OntoProtein [66] and InstructProtein [59] incorporate external knowledge graphs into protein pretraining. ProtST [61] enhances protein pretraining and understanding by biomedical texts with BLM, while Prot2Text [1] has achieved function prediction in free text format from graph and sequential inputs. ProLLaMA [33] utilizes the pretrained LLaMA2 to perform continual learning on protein language. Moreover, frameworks like BioT5 [42] and BioT5+ [40] have been proposed to capture the underlying relations and characteristics of different bio-entities. These developments not only underscore the potential for bridging natural language and protein language but also provide valuable insights for future multi-modal learning research.

**Protein active sites identification** The active site[1] of a protein is a crucial region where it interacts with other molecules. It is typically composed of amino acid residues, and the arrangement determines the protein's structure and function. Thus, protein active sites identification is essential for understanding the protein's role within organisms. Over a decade ago, various methods were employed for this purpose, including statistics-based approaches [46], protein surface modelling [15], and straightforward machine learning techniques [25] such as Random Forest and Support Vector Machine [64]. However, these methods were either challenging to implement or lack effectiveness. Recently, deep learning has been leveraged to predict the ligand binding sites on proteins. For instance, DeepSurf [37] and CrossBind [23] utilize 3D voxelized grids and point clouds, respectively, to generate volumetric protein representations. Meanwhile, GraphBind [60], ScanNet [55], and DeepProSite [13] integrate both primary sequences and tertiary structures to recognize amino acids in the binding site region. However, although active sites are directly involved in the activity of a protein and play a significant role in drug design, enzyme engineering, *etc.*, they are relatively less studied.

## 3 Methods

In this section, we first give a formal definition to the multi-modal active sites identification task in Section 3.1. Then, we present our MMSite framework in Section 3.2, which comprises attributes reconstruction, feature extraction, multi-modal alignment and fusion.

---

[1]Following [6], *active site* denotes the amino acid(s) directly involved in the activity of an enzyme.

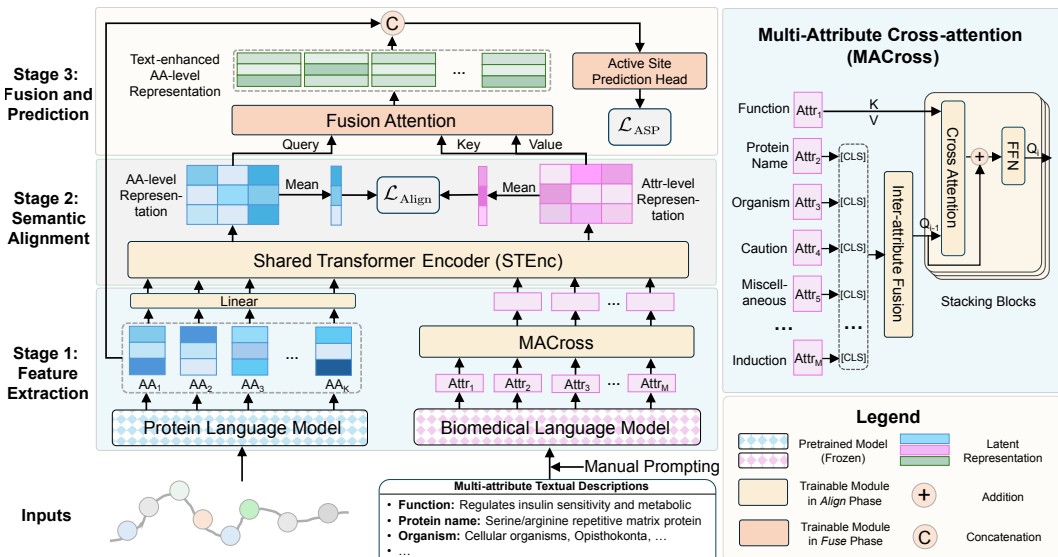

Figure 2: Overview of the MMSite framework. MMSite takes paired sequences and multi-attribute textual descriptions as inputs, using PLM and BLM to extract features in Stage 1. For the text modality, manual prompting and the MACross Module process the multi-attribute descriptions. A "First Align, Then Fuse" strategy is then employed to align and fuse both modalities. Specifically, in Stage 2, a Shared Transformer Encoder and soft-label alignment align the dual modalities. In Stage 3, Fusion Attention and a skip concatenation strategy are used to predict active sites, with only the modules in Stage 3 being trainable. Note: During inference, the missing text modality is generated by an agent model and directly input into the Shared Transformer Encoder, bypassing the need to process through MACross, as it is not multi-attribute.

## 3.1 Problem definition

Here we formulate our task of predicting active sites in proteins using both protein sequences and multi-attribute textual descriptions. We define the dataset composition in our ProTAD as $\mathcal{D} = \{\mathcal{S}, \mathcal{T}\}$, where $\mathcal{S} = \{S_i\}_{i=1}^N$ represents the primary sequences and $\mathcal{T} = \{T_i\}_{i=1}^N$ denotes the textual attributes associated with each protein, with $N$ being the total number of entries in the dataset. For the $i$-th protein, $S_i$ is a sequence of $k_i$ amino acids, denoted as $S_i = \{\mathrm{AA}_i^1, \mathrm{AA}_i^2, \cdots, \mathrm{AA}_i^{k_i}\}$. Each $T_i$ is a collection of $M$ textual descriptions that depict protein from different perspectives. These descriptions are structured as pairs consisting of an attribute name $\mathrm{t}^n$ and attribute content $\mathrm{t}^c$ in raw data, specifically, $T_i = \{(\mathrm{t}_{i,j}^n, \mathrm{t}_{i,j}^c)\}_{j=1}^M$. As for the annotation target at the token-level, $\boldsymbol{y}_i = \{y_{i,j}\}_{j=1}^{k_i}$ indicates whether each amino acid in the sequence is an active site, where $y_{i,j} \in \{0, 1\}$. During the training stage, both modalities $\mathcal{S}$ and $\mathcal{T}$ are employed to develop our model. The likelihood that each amino acid is an active site is predicted using only the protein sequence $\mathcal{S}$ as an input during the inference stage.

## 3.2 Contrastive learning-based alignment and fusion

**Attribute description reconstruction with prompt** In our raw data, textual descriptions are structured with each protein having serveral distinct attribute descriptions in the form $(\mathrm{t}^n, \mathrm{t}^c)$, formatted as ATTRIBUTE_NAME:ATTRIBUTE_CONTENT, such as:

- Protein Name: Parkinson disease protein 7 homolog;
- Taxonomic Lineage: Cellular organisms, Opisthokonta, Eumetazoa;
- Function: Multifunctional protein with controversial molecular function which plays an important role in cell protection against oxidative stress.

In order to handle this tabular-like form of data, enhancing the linkage between the content and its corresponding attribute name is important. We designed manual *prompts* for each attribute to

reconstruct the tabular text pairs into full sentences, thus $\tilde{T} = \{\tilde{\mathrm{t}}_j\}_{j=1}^M$. Consequently, the examples above are reconstructed as follows:

- *The name of protein is* Parkinson disease protein 7 homolog.
- *The taxonomic lineage of this protein includes* Cellular organisms, Opisthokonta, Metazoa.
- *The function of this protein includes* Multifunctional protein with controversial molecular function which plays an important role in cell protection against oxidative stress.

The words in *italics* are the prompts that not only aid in reconstructing free-formed texts, but also provide a more relevant and comprehensive description of each protein.

**Modality feature extraction**    During the training stage, the pretrained PLM $f_\phi$ and BLM $f_\psi$ are used to initialize the representations of protein sequence $f_\phi(S)$ and the textual descriptions $f_\psi(\tilde{T})$, respectively. We freeze the weights of both the PLM and BLM due to the expensive computational cost and design subsequent learnable modules to adapt these models for our task.

Specifically, $M$ reconstructed textual descriptions are fed into BLM:

$$f_\psi(\tilde{T}) = \{f_\psi(\tilde{\mathrm{t}}_1), f_\psi(\tilde{\mathrm{t}}_2), \cdots, f_\psi(\tilde{\mathrm{t}}_M)\}, \tag{1}$$

where $f_\psi(\tilde{\mathrm{t}}_i) \in \mathbb{R}^{l_i \times d}$, and the truncated length of tokenized sequence $l_i$ is not same for different attributes for saving computational resources (details can be found in Appendix B.2). Given the hierarchical characteristic of textual descriptions, we design a *Multi-Attribute Cross-attention* (MACross) module to capture the relationship among attributes. Intuitively, the Function attribute is of the greatest importance due to the rich information it carries. Thus, we first incorporate the left $M - 1$ attributes by [CLS] token [9] with inter-attribute attention to obtain $\boldsymbol{x}_{-F}^{\mathrm{t}}$:

$$\boldsymbol{x}_{-F}^{\mathrm{t}} = \mathrm{Attention}(\mathrm{Concat}(\{f_\psi(\tilde{\mathrm{t}}_i)_{\texttt{[CLS]}}|1 \leq i \leq M, \mathrm{t}_i^n \neq \texttt{Function}\})), \tag{2}$$

while $\boldsymbol{x}_F^{\mathrm{t}} := f_\psi(\tilde{\mathrm{t}}_i)$ where $\mathrm{t}_i^n = \texttt{Function}$. Then we employ multi-layer cross-attention [57] to query the information of $\boldsymbol{x}_F$ by $\boldsymbol{x}_{-F}$:

$$\mathrm{CrossAttention}(\boldsymbol{x}_{-F}^{\mathrm{t}}, \boldsymbol{x}_F^{\mathrm{t}}, \boldsymbol{x}_F^{\mathrm{t}}) = \mathrm{Softmax}(\frac{\mathbf{Q}_{-F}\mathbf{K}_F^\top}{\sqrt{d}})\mathbf{V}_F, \tag{3}$$

$$\mathbf{Q}_{-F} = \mathbf{W}^Q \boldsymbol{x}_{-F}^{\mathrm{t}}; \mathbf{K}_F = \mathbf{W}^K \boldsymbol{x}_F^{\mathrm{t}}; \mathbf{V}_F = \mathbf{W}^V \boldsymbol{x}_F^{\mathrm{t}}, \tag{4}$$

where $\mathbf{W}^Q$, $\mathbf{W}^K$ and $\mathbf{W}^V$ refer to learnable transformation matrices, and $d$ refers to the dimension of each attention head. Similar to [28], we employ a residual connection by a fully connected feed-forward network (FFN) as shown in Figure 2.

**Cross-modal soft-label alignment**    In order to utilize the complementary knowledge of amino acid sequence and textual descriptions to improve the performance and robustness. Inspired by [29], we employ the "First Align, Then Fuse" strategy to fuse two aligned modalities to predict the amino acid-level target (*i.e.*, the active sites) of protein. Assume that the output of the sequence and textual description after feature extraction are $\boldsymbol{z}^{\mathrm{s}}$ and $\boldsymbol{z}^{\mathrm{t}}$, respectively, although both of them point to the same protein, the divergence still exists. Therefore, firstly we use a *Shared Transformer Encoder* (STEnc, consisting of multiple Transformer encoders) to early-map $\boldsymbol{z}^{\mathrm{s}}$ and $\boldsymbol{z}^{\mathrm{t}}$:

$$\tilde{\boldsymbol{z}}^{\mathrm{s}} = \mathrm{STEnc}(\boldsymbol{z}^{\mathrm{s}}); \tilde{\boldsymbol{z}}^{\mathrm{t}} = \mathrm{STEnc}(\boldsymbol{z}^{\mathrm{t}}). \tag{5}$$

In order to align paired sequence and textual descriptions, it is common practice to use contrastive learning based on InfoNCE loss [38]. This method "pulls close" the paired samples ("positive pair") as "pushes away" the unpaired samples ("negative pair") in the high-dimension space by maximising mutual information between two modalities in self-supervised manner. However, in our case there may be a potential semantic association between unpaired sequence and textual descriptions in the same batch due to the principle "*Similar protein sequences give rise to similar structures and functions*" in biology (Appendix C). Inspired by [20], we adopt cross-modal soft-label alignment to make one-hot label continuous.[2] Specifically, the cosine similarity between $\tilde{\boldsymbol{z}}_i^{\mathrm{s}}$ and $\tilde{\boldsymbol{z}}_j^{\mathrm{t}}$ is denoted as $s_{ij}^{\mathrm{s2t}}$, and the cosine similarity between $\tilde{\boldsymbol{z}}_i^{\mathrm{s}}$ and $\tilde{\boldsymbol{z}}_j^{\mathrm{s}}$ [3] is denoted as $r_{ij}^{\mathrm{s2s}}$, and $r_{ij}^{\mathrm{t2t}}$ is defined in the

---

[2] We don't adopt the uni-modal alignment because it's not applicable to our task.

[3] We calculate the mean results of $\tilde{\boldsymbol{z}}^{\mathrm{s}}$ and $\tilde{\boldsymbol{z}}^{\mathrm{t}}$ in length dimension as the representations of two modalities, respectively, in Equation 6 and 7.

same way. $P_{ij}^{\text{s2s}}$ represents the semantic consistency within the same modality which is calculated in a *softmax-like* manner:

$$P_{ij}^{\text{s2s}} = \frac{\exp(r_{ij}^{\text{s2s}})}{\sum_{k=1}^{|\mathcal{B}|} \exp(r_{ik}^{\text{s2s}})}, \tag{6}$$

where $|\mathcal{B}|$ is batch size. Actual distribution $Q_{ij}^{\text{s2t}}$ represents the probability that $\tilde{z}_i^{\text{s}}$ matches $\tilde{z}_j^{\text{t}}$, which is hoped to align with $P_{ij}^{\text{s2s}}$:

$$Q_{ij}^{\text{s2t}} = \frac{\exp(s_{ij}^{\text{s2t}}/\tau)}{\sum_{k=1}^{|\mathcal{B}|} \exp(s_{ik}^{\text{s2t}}/\tau)}, \tag{7}$$

where $\tau$ is temperature. Thus, the loss function in the *Align* phase can be calculated as:

$$\mathcal{L}_{\text{Align}} = \frac{1}{2|\mathcal{B}|} \sum_{i=1}^{|\mathcal{B}|} (D_{KL}(P_i^{\text{s2s}} \| Q_i^{\text{s2t}}) + D_{KL}(P_i^{\text{t2t}} \| Q_i^{\text{t2s}})). \tag{8}$$

We note that the parameters of the sequence branches are frozen, resulting in the alignment of the textual feature space to that of the sequence feature space.

**Multi-modal fusion and active sites identification** In the *Fuse* phase, a multi-head cross-attention strategy is utilized again to integrate the protein and text modalities, where $\tilde{z}^{\text{s}}$ serves as "query" while $\tilde{z}^{\text{t}}$ serves as both "key" and "value". This setup enables the network to develop a comprehensive representation of queried text modality by sequence. Consequently, the model not only encodes protein-related knowledge but also retains insights derived from textual data, fostering a holistic comprehension. Subsequently, the unmapped $z^{\text{s}}$ and the features deviated by *Fusion Attention* are concatenated before the prediction layer. Finally, the model employs Cross Entropy Loss, denoted as $\mathcal{L}_{\text{ASP}}$, to predict active sites.

Although during the training stage, both sequence and text modalities are used to predict active sites, in practice, when dealing with newly discovered proteins, we might lack its high-quality textual annotations. To address this issue, we employ a state-of-the-art biomedical text generation method, such as Prot2Text [1], as an agent to complete the missing text modality. Some examples of generated texts are shown in Appendix B.4.

## 4 Experiments

### 4.1 Experiment setups

**Dataset description** To obtain comprehensive and accurate protein-text data, we build the ProTAD with 570,830 samples from Swiss-Prot in UniProt[4] [6] after data cleaning and filtering. ProTAD includes the amino acid sequence and textual descriptions for each sample, covering 17 attributes such as `Protein Name`, `Organism`, `Function`, `Caution`, *etc.* as described in Appendix A.1. These descriptions are rigorously checked so that the proteins can be accurately described. Due to some proteins lacking certain attribute annotations, we select those pairs of samples with at least six attributes (of which the `Function` attribute is required) in our experiments. To ensure a fair comparison with other methods when conducting experiments in Section 4.2.1, we filter the proteins those could obtain tertiary structures in AlphaFold DB [24] [56]. In order to prevent the potential data leakage, following CLEAN [65], we cluster the data using MMseqs2 [35] with different sequence identity thresholds (in our settings they are 10%, 30%, 50%, 70%, and 90%) to avoid the test sequences from being too similar to the training data. After that, we develop a *cluster-guarantee* approach and employ *k-selected* strategy to construct an $8:1:1$ split dataset. The detailed preparation process can be found in Appendix A.2.

**Implementation details** Our implementation is based on PyTorch version 1.13.1, and models are trained using a single NVIDIA GeForce RTX 4090 GPU with 24GB of memory. The MMSite model in Table 1 requires approximately 7 hours to train. We retain the original feature dimensions of each PLM encoder and BLM encoder to preserve more inherent information, *e.g.*, 1280 dimensions for ESM-2-650M and 768 dimensions for PubMedBERT-abs [16]. The maximum sequence length is

---

[4]We collected data in UniProt (`https://www.uniprot.org`), deposited before March 11th, 2024.

Table 1: Comparison on the dataset with clustering threshold at 10% compared with other 21 PRL models. All results are reported as mean$(\pm 2\sigma)$. *Abbr.*, Seq.: Sequence; Struct.: Structure.

| Input[†] | Method | Version | $F_{max}$ ↑ | AUPRC ↑ | MCC ↑ | OS ↑ | FPR ↓ |
|---|---|---|---|---|---|---|---|
| Seq. | ESM | 1b [45] | 0.7052$(\pm 0.02)$ | 0.8452$(\pm 0.02)$ | 0.7123$(\pm 0.02)$ | 0.7211$(\pm 0.04)$ | 0.2758$(\pm 0.01)$ |
| | | 1v [34] | 0.6306$(\pm 0.03)$ | 0.7975$(\pm 0.02)$ | 0.6382$(\pm 0.03)$ | 0.6398$(\pm 0.03)$ | 0.3388$(\pm 0.02)$ |
| | | 2-650M [30] | 0.6517$(\pm 0.04)$ | 0.8230$(\pm 0.04)$ | 0.6596$(\pm 0.04)$ | 0.6652$(\pm 0.02)$ | 0.3240$(\pm 0.05)$ |
| | ProtT5 [12] | BFD | 0.4156$(\pm 0.05)$ | 0.6773$(\pm 0.03)$ | 0.4217$(\pm 0.05)$ | 0.4130$(\pm 0.05)$ | 0.5509$(\pm 0.05)$ |
| | | UniRef | 0.4696$(\pm 0.04)$ | 0.7119$(\pm 0.02)$ | 0.4767$(\pm 0.04)$ | 0.4652$(\pm 0.04)$ | 0.4919$(\pm 0.04)$ |
| | ProtBert [12] | BFD | 0.5610$(\pm 0.02)$ | 0.7524$(\pm 0.02)$ | 0.5715$(\pm 0.02)$ | 0.5865$(\pm 0.04)$ | 0.4115$(\pm 0.02)$ |
| | | UniRef | 0.4817$(\pm 0.02)$ | 0.6992$(\pm 0.01)$ | 0.4896$(\pm 0.02)$ | 0.4915$(\pm 0.01)$ | 0.4871$(\pm 0.03)$ |
| | ProtAlbert [12] | | 0.6033$(\pm 0.03)$ | 0.7519$(\pm 0.02)$ | 0.6121$(\pm 0.03)$ | 0.6149$(\pm 0.03)$ | 0.3636$(\pm 0.01)$ |
| | ProtXLNet [12] | | 0.0345$(\pm 0.00)$ | 0.0952$(\pm 0.02)$ | 0.0409$(\pm 0.00)$ | 0.0772$(\pm 0.01)$ | 0.9233$(\pm 0.00)$ |
| | ProtElectra [12] | | 0.5636$(\pm 0.02)$ | 0.7630$(\pm 0.02)$ | 0.5732$(\pm 0.02)$ | 0.5793$(\pm 0.04)$ | 0.4041$(\pm 0.01)$ |
| | PETA [52] | deep_base | 0.6533$(\pm 0.02)$ | 0.7994$(\pm 0.01)$ | 0.6603$(\pm 0.02)$ | 0.6529$(\pm 0.02)$ | 0.3134$(\pm 0.02)$ |
| | S-PLM [58] | | 0.7262$(\pm 0.02)$ | 0.8712$(\pm 0.01)$ | 0.7337$(\pm 0.02)$ | 0.7322$(\pm 0.03)$ | 0.2452$(\pm 0.02)$ |
| | TAPE [44] | | 0.3560$(\pm 0.02)$ | 0.5413$(\pm 0.01)$ | 0.3622$(\pm 0.02)$ | 0.3523$(\pm 0.02)$ | 0.6096$(\pm 0.02)$ |
| Seq. & Struct. | MIF [62] | MIF | 0.1379$(\pm 0.02)$ | 0.3470$(\pm 0.02)$ | 0.1393$(\pm 0.02)$ | 0.1346$(\pm 0.02)$ | 0.8524$(\pm 0.03)$ |
| | | MIF-ST | 0.1033$(\pm 0.02)$ | 0.2883$(\pm 0.03)$ | 0.1034$(\pm 0.02)$ | 0.1030$(\pm 0.02)$ | 0.8958$(\pm 0.02)$ |
| | PST [3] | t33 | 0.6574$(\pm 0.01)$ | 0.8139$(\pm 0.01)$ | 0.6648$(\pm 0.01)$ | 0.6719$(\pm 0.01)$ | 0.3219$(\pm 0.01)$ |
| | | t33_so | 0.6708$(\pm 0.02)$ | 0.8266$(\pm 0.03)$ | 0.6793$(\pm 0.02)$ | 0.6891$(\pm 0.03)$ | 0.3079$(\pm 0.01)$ |
| Seq. & Text | ProtST [61] w/ retrain | ESM-1b | 0.4036$(\pm 0.03)$ | 0.6762$(\pm 0.02)$ | 0.4144$(\pm 0.02)$ | 0.4297$(\pm 0.03)$ | 0.5663$(\pm 0.02)$ |
| | | ESM-2 | 0.1865$(\pm 0.01)$ | 0.4220$(\pm 0.03)$ | 0.1918$(\pm 0.02)$ | 0.1872$(\pm 0.05)$ | 0.7897$(\pm 0.01)$ |
| | ProtST w/o retrain | ESM-1b | 0.4632$(\pm 0.05)$ | 0.7040$(\pm 0.02)$ | 0.4722$(\pm 0.05)$ | 0.4779$(\pm 0.05)$ | 0.5030$(\pm 0.05)$ |
| | | ESM-2 | 0.5483$(\pm 0.02)$ | 0.7716$(\pm 0.01)$ | 0.5562$(\pm 0.02)$ | 0.5613$(\pm 0.01)$ | 0.4239$(\pm 0.02)$ |
| | **MMSite ‡** | | **0.8250**$(\pm 0.02)$ | **0.8909**$(\pm 0.01)$ | **0.8319**$(\pm 0.02)$ | **0.8549**$(\pm 0.02)$ | **0.1689**$(\pm 0.02)$ |

† This column refers to the modality input in the inference stage.
‡ We report the performance using ESM-1b and PubMedBERT-abs as the PLM and BLM encoders.

512, and extra amino acids will be removed. Details on the truncated length for each attribute can be found in Appendix B.2. We set the hyperparameter $\tau$ to 0.8. In MACross, we use 2-layer Transformer encoder to extract inter-attribute relations, with the number of cross-attention blocks set to 4. We also adopt 4-layer 8-head Transformer in STEnc. Before the final MLP predictor, an additional 2-layer attention mechanism integrates the original sequence modality with the fused feature. The dropout rates of all above components is consistently set at 0.1. The model undergoes a 15-epoch *Align* phase and a 50-epoch *Fuse* phase, using the Adam optimizer with a learning rate of 5e-5. We implement a warm-up phase comprising $\frac{1}{10}$ of the total steps, followed by a cosine annealing scheduler for the remaining steps. The batch size is set to 24 for both training and inference stages. We consider those sites as predicted active sites whose result, after the Sigmoid function of the MLP predictor output, is greater than 0.5.

## 4.2 Protein active sites identification

### 4.2.1 Comparison with baselines

**Settings** To compare with existing methods, we select 21 state-of-the-art PRL models as baselines. Their original weights are frozen, and residue-level features are obtained followed by 4-layer Transformer for prediction. Some of them utilize not only sequences but also combinations of sequences with structures, and sequences with text. Specifically, for models like MIF and PST, we obtain tertiary structures for each protein from AlphaFold DB. For ProtST, we perform comparisons w/ and w/o retraining on ProTAD. For MMSite, Prot2Text [1] serves as the agent model in the inference stage. Evaluation metrics include token-level $F_{max}$, AUPRC, and MCC, following the implementation described in [21], as well as region-level OS (Overlap Score) and FPR (False Positive Rate) as defined in [8]. We save the checkpoint at the epoch with the best $F_{max}$ in validation set. Results are reported in Table 1 for the dataset with clustering threshold at 10%, using several different seeds, where ESM-1b and PubMedBERT-abs are used as the PLM and BLM encoders to initialize features. Some visualisation results are presented in Figure 3 and Appendix F.

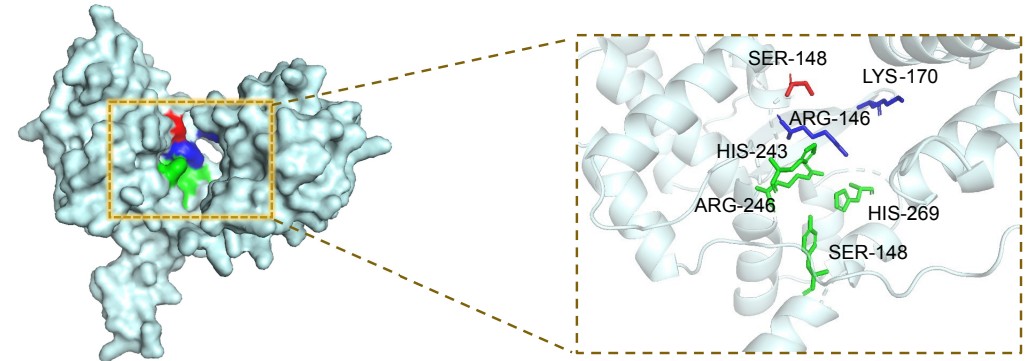

Figure 3: Visualisation of an example of active site identification result for the protein *Tyrosine recombinase XerC* (UniProt ID: Q039E1). The **palecyan** surface/sticks (residues) represent the background, while the **green**, **blue**, and **red** surface/sticks (residues) indicate the correctly predicted sites, unpredicted sites, and incorrectly predicted sites, respectively.

**Results and discussions**   The results demonstrate that MMSite outperforms individual models that only use residue sequences as input, achieving state-of-the-art performance across all metrics. Among the comparisons, S-PLM, which incorporates contrastive learning between sequences and structures during pretraining, performs slightly better than other Seq.-input methods, showing the potential of incorporating structural data. Nonetheless, MMSite still outperforms the model with Seq. & Struct. as input. Regarding ProtST w/ and w/o retrain, limited utilization of textual information in retraining process and reduced data volume for downstream application lead to decreased performance.

### 4.2.2   BLM enhances PLM's performance

Evolutionary information is crucial in the identification of function sites, as these sites often preserve conserved patterns and structural motifs across species or among homologous proteins. In Table 2, we compare three state-of-the-art evolutionary-scale models (*e.g.*, ESM-1b, ESM-1v, and ESM-2-650M) that are used as encoders for protein sequences. Additionally, BLMs such as PubMedBERT-abs and PubMedBERT-full serve as biological text encoders. It is evident that the utilization of BLM has resulted in significant enhancements of the performance of PLM.

Table 2: Performance improvement with the addition of BLM compared to using PLM as input only.

| Method | $F_{max}$ ↑ | AUPRC ↑ | MCC ↑ | OS ↑ | FPR ↓ |
|---|---|---|---|---|---|
| ESM-1b | 0.7050 | 0.8443 | 0.7117 | 0.7127 | 0.2705 |
| **+MMSite-abs** | ↑ 0.120 | ↑ 0.047 | ↑ 0.120 | ↑ 0.142 | ↓ 0.102 |
| **+MMSite-full** | ↑ 0.105 | ↑ 0.044 | ↑ 0.107 | ↑ 0.145 | ↓ 0.076 |
| ESM-1v | 0.6267 | 0.8018 | 0.6340 | 0.6351 | 0.3431 |
| **+MMSite-abs** | ↑ 0.160 | ↑ 0.069 | ↑ 0.159 | ↑ 0.164 | ↓ 0.149 |
| **+MMSite-full** | ↑ 0.172 | ↑ 0.078 | ↑ 0.172 | ↑ 0.184 | ↓ 0.156 |
| ESM-2-650M | 0.6402 | 0.8068 | 0.6479 | 0.6607 | 0.3434 |
| **+MMSite-abs** | ↑ 0.156 | ↑ 0.072 | ↑ 0.157 | ↑ 0.175 | ↓ 0.138 |
| **+MMSite-full** | ↑ 0.162 | ↑ 0.075 | ↑ 0.161 | ↑ 0.169 | ↓ 0.152 |

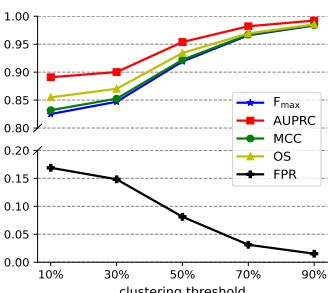

Figure 4: Impact of clustering threshold on model performance.

### 4.3   Ablation study

### 4.3.1   Impact of different clustering thresholds

In order to study the impact of identity thresholds on clustering using MMseqs2 during data partition, we set thresholds at 10%, 30%, 50%, 70%, and 90% for comparison, and the results are shown in Figure 4, where ESM-1b and PubMedBERT-abs serve as the PLM and BLM encoders respectively. Detailed quantitative comparisons for the cases of 30% and 50% are provided in Appendix D.4. It is clearly that with the increase of clustering threshold, all the performances of each metric are improved. Especially when the threshold is changed from 30% to 50%, the improvement of the model is especially obvious, and when it reaches 90%, the performance approaches near perfection.

### 4.3.2 Effectiveness of components

To figure out the contribution of each component within the MMSite framework to overall model performance, we conduct ablation experiments for each of them. The results are presented in Table 3, where "Seq-M" and "Text-M" refer to the sequence modality and the text modality, respectively. It can be found that both STEnc and MACross contribute to the performance improvement, and the *Align* mechanism helps more obviously by aligning the text modality closer to the sequence modality. Moreover, the importance of the text modality is evident in the last row, which shows an average decline of 0.105 across all metrics when the text modality is removed, compared to MMSite.

Table 3: Evaluation of the effectiveness of each component in MMSite.

| Seq-M | Text-M | Align | MACross | STEnc | $F_{max}\uparrow$ | AUPRC $\uparrow$ | MCC $\uparrow$ | OS $\uparrow$ | FPR $\downarrow$ |
|---|---|---|---|---|---|---|---|---|---|
| ✓ | ✓ | ✓ | ✓ | ✓ | **0.8250** | **0.8909** | **0.8319** | **0.8549** | **0.1689** |
| ✓ | ✓ | ✓ | ✓ | | 0.8021 | 0.8819 | 0.8071 | 0.8027 | 0.1738 |
| ✓ | ✓ | ✓ | | ✓ | 0.8152 | 0.8908 | 0.8214 | 0.8379 | 0.1757 |
| ✓ | ✓ | ✓ | | | 0.8037 | 0.8850 | 0.8105 | 0.8241 | 0.1847 |
| ✓ | ✓ | | | | 0.7911 | 0.8710 | 0.7980 | 0.8150 | 0.1978 |
| ✓ | | | | | 0.7052 | 0.8452 | 0.7123 | 0.7211 | 0.2758 |

### 4.3.3 Single-stage *vs.* two-stage

Our framework employs a two-stage training strategy, "First Align, Then Fuse", and we also compare it with a single-stage strategy "Align While Fusing". The total loss for the single-stage strategy is calculated as $\mathcal{L}_{total} = \mathcal{L}_{Align} + \alpha\mathcal{L}_{ASP}$, where $\alpha$ is set to the best performing 0.7 after many attempts. The comparative results are presented in Table 4. It is clear that the two-stage strategy outperforms the single-stage approach, which is more challenging due to its multi-objective nature and the potential conflicts between different objectives.

Table 4: Comparison between the single-stage and the two-stage strategy.

| PLM | Strategy | PubMedBERT-abs | | | | | PubMedBERT-full | | | | |
|---|---|---|---|---|---|---|---|---|---|---|---|
| | | $F_{max}\uparrow$ | AUPRC $\uparrow$ | MCC $\uparrow$ | OS $\uparrow$ | FPR $\downarrow$ | $F_{max}\uparrow$ | AUPRC $\uparrow$ | MCC $\uparrow$ | OS $\uparrow$ | FPR $\downarrow$ |
| ESM-1b | Single | 0.8086 | 0.8772 | 0.8158 | 0.8329 | 0.1798 | 0.8055 | 0.8766 | 0.8121 | 0.8253 | **0.1818** |
| | Two | **0.8250** | **0.8909** | **0.8319** | **0.8549** | **0.1689** | **0.8099** | **0.8882** | **0.8183** | **0.8574** | 0.1950 |
| ESM-1v | Single | 0.7369 | 0.8525 | 0.7440 | 0.7576 | 0.2480 | 0.7713 | 0.8589 | 0.7780 | 0.7924 | 0.2155 |
| | Two | **0.7864** | **0.8705** | **0.7933** | **0.7987** | **0.1942** | **0.7988** | **0.8795** | **0.8058** | **0.8194** | **0.1871** |
| ESM-2 -650M | Single | 0.7522 | 0.8572 | 0.7591 | 0.7664 | 0.2296 | 0.7603 | 0.8638 | 0.7669 | 0.7677 | 0.2171 |
| | Two | **0.7965** | **0.8789** | **0.8046** | **0.8358** | **0.2052** | **0.8018** | **0.8814** | **0.8091** | **0.8294** | **0.1916** |

### 4.3.4 Ablation of the attribute selection

MACross module is designed based on that the `Function` attribute is the most relevant attribute for predicting active sites and contains the richest information. The additional attributes, although seemingly insignificant, also contribute to improve active site predictions. Table 5 below shows the performance comparison between using only the `Function` attribute and using all attributes, demonstrating that incorporating all attributes leads to better performance.

Table 5: Performance comparison between using all attributes and only `Function` attribute.

| Textual Description | $F_{max}\uparrow$ | AUPRC $\uparrow$ | MCC $\uparrow$ | OS $\uparrow$ | FPR $\downarrow$ |
|---|---|---|---|---|---|
| All attributes | **0.8250** | **0.8909** | **0.8319** | **0.8549** | **0.1689** |
| Only `Function` attribute | 0.8152 | 0.8866 | 0.8231 | 0.8471 | 0.1764 |

### 4.3.5 Ablation of manual prompting

To evaluate the effectiveness of manual prompting, we conducted experiments to test the impact of removing manual prompts in Table 6. The results show that manual prompting improves performance on most metrics. We believe this is because: (1) Complete sentences provide richer context for the

BERT-based BLM; (2) It reduces ambiguity in attribute meanings; (3) It aligns better with BLM's pretraining, leveraging its knowledge more effectively.

Table 6: Ablation experiment of manual prompting.

| Method | $F_{max}$ ↑ | AUPRC ↑ | MCC ↑ | OS ↑ | FPR ↓ |
|---|---|---|---|---|---|
| w manual prompting | **0.8250** | 0.8909 | **0.8319** | **0.8549** | **0.1689** |
| w/o manual prompting | 0.8157 | **0.8911** | 0.8221 | 0.8460 | 0.1793 |

### 4.3.6 Other ablation studies

In MACross, the `Function` and the remaining 16 attributes are utilized as K & V, and Q respectively in Cross Attention to query $x_F$. We attempt to swap their positions for comparison (*i.e.*, $x_F$ serves as "query", $x_{-F}$ servers as "key" and "value"). Additionally, we also try to replace soft-label alignment with hard-label alignment (similar to InfoNCE). The results of their average performance on token-level and region-level are shown in Table 7. To compare the results, we replace FPR with $1 - $ FPR when calculating Avg. (region). MMSite performs best compared to the other two scenarios. We also investigated the impact of varying the hyperparameter $\tau$ in soft-label alignment from 0.2 to 2.0 in Figure 5, because it directly determines the information entropy of the target distribution $Q$ in the *Align* phase. The results shows that the model is relative optimal in both token-level and region-level metrics when $\tau$ is set to 0.8.

Table 7: Comparison between MMSite and other two scenarios.

| Method | Avg. (token) | Avg. (region) |
|---|---|---|
| **MMSite** | **0.8493** | **0.8430** |
| Func. as Q | 0.8394 | 0.8279 |
| Hard align | 0.8334 | 0.8221 |

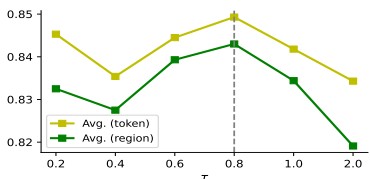

Figure 5: Performance on different $\tau$.

### 4.4 Temporal-based evaluation

To more accurately reflect real-world scenarios in scientific applications, we conducted an extra experiment simulating the discovery of new proteins. We collected data from UniProt database recorded after March 11, 2024 as newly discovered proteins (115 samples) to evaluate the model's performance. The results in Table 8 show that MMSite maintains well performance even on newly discovered proteins.

Table 8: Performance evaluation on the newly discovered proteins.

| Test Data | $F_{max}$ | AUPRC | MCC | OS | FPR |
|---|---|---|---|---|---|
| Newly discovered proteins | 0.8432 | 0.8865 | 0.8460 | 0.8465 | 0.1420 |

Due to space limitations, further experiments results on performance comparisons, text quality, protein sequence input, inference performance, and related aspects are included in the Appendix D.

## 5 Conclusion

In this work, we build the ProTAD dataset that contains detail textual descriptions of proteins, and propose the MMSite framework to identify the active sites in proteins, which is crucial for understanding proteins in residue-level, designing new drugs and so on. MMSite takes both sequence and text as input in training stage, and adopts "First Align, Then Fuse" strategy to align the text representation to the sequence, enhancing PLM's performance in active sites identification. We adopt manual prompting and a designed MACross module to handle the multi-attribute descriptions, and adopt soft-label alignment in the *Align* phase. Extensive experimental validations have been conducted to evaluate the effectiveness of our method, showing the potential of multi-modal learning in computational biology. From experience, the method works best when ESM-1b and PubMedBERT-abs are chosen as the PLM and BLM, Prot2Text is used as the agent model. We also discussed the limitations and broader impacts in Appendix E.

## Acknowledgments and Disclosure of Funding

This work is supported by the National Key Research and Development Program of China (2023YFC2705700), the National Natural Science Foundation of China (Grant No. 62225113, 62272354, 62276195 and U23A20318), the Science and Technology Major Project of Hubei Province under Grant 2024BAB046, and the Innovative Research Group Project of Hubei Province under Grant 2024AFA017. The numerical calculations in this paper have been done on the supercomputing system in the Supercomputing Center of Wuhan University.

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

# A  Dataset details

## A.1  Description

Taking the protein sequence P05117 (the UniProt ID) as an example, below are the raw textual descriptions of its attributes from our ProTAD dataset, which cover 17 different attributes such as `Protein Name`, `Organism`, `Taxonomic Lineage`, *etc.* These descriptions comprehensively and accurately reflect the characteristics of the protein. It is worth noting that some proteins in our dataset have missing attributes, such as `Caution`, `Allergenic Properties`, `Pharmaceutical use` and `Involvement in Disease` for P05117.

---

**Raw Textual Descriptions of P05117**

- `Protein Name`: Polygalacturonase-2 (PG) (PG-2A) (PG-2B) (Pectinase)
- `Organism`: Solanum lycopersicum (Tomato) (Lycopersicon esculentum)
- `Taxonomic Lineage`: cellular organisms (no rank), Eukaryota (superkingdom), Viridiplantae (kingdom), Streptophyta (phylum), Streptophytina (subphylum), Embryophyta (no rank), Tracheophyta (no rank), Euphyllophyta (no rank), Spermatophyta (no rank), Magnoliopsida (class), Mesangiospermae (no rank), eudicotyledons (no rank), Gunneridae (no rank), Pentapetalae (no rank), asterids (no rank), lamiids (no rank), Solanales (order), Solanaceae (family), Solanoideae (subfamily), Solaneae (tribe), Solanum (genus), Solanum subgen. Lycopersicon (subgenus)
- `Function`: Catalytic subunit of the polygalacturonase isozyme 1 and 2 (PG1 and PG2). Acts in concert with the pectinesterase, in the ripening process. Is involved in cell wall metabolism, specifically in polyuronide degradation. The depolymerization and solubilization of cell wall polyuronides mediated by PG2 during ripening seems to be limited by the beta subunit GP1, probably by recruiting PG2 to form PG1.
- `Caution`: **nan**.
- `Miscellaneous`: To avoid liquid rheology of tomato juice, temperature and pressure can be increased to inactivate selectively PG2 during the process.
- `Subunit Structure`: Monomer PG2 (isoenzymes PG2A and PG2B). Also forms heterodimers called polygalacturonase 1 (PG1) with the beta subunit GP1.
- `Induction`: By ethylene.
- `Tissue Specificity`: Expressed only in ripening fruits (at protein level).
- `Developmental Stage`: PG1 appears when fruits start to be coloured. When fruits are orange, both PG2 and PG1 are present. In fully ripe fruit, mostly PG2 is expressed.
- `Allergenic Properties`: **nan**
- `Biotechnological Use`: The effect of PG can be neutralized by introducing an antisense PG gene by genetic manipulation. The Flavr Savr tomato produced by Calgene (Monsanto) in such a manner has a longer shelf life due to delayed ripening.
- `Pharmaceutical Use`: **nan**
- `Involvement in Disease`: **nan**
- `Subcellular Location`: Secreted, extracellular space, apoplast. Secreted, cell wall.
- `Post-translational Modification`: N-glycosylated. PG2B isozyme has a greater degree of glycosylation than PG2A.
- `Sequence Similarities`: Belongs to the glycosyl hydrolase 28 family.

---

To adapt to the model input and simultaneously establish the relationship between the attribute names and their content, we manually designed a unique prompt for each attribute to bridge the relationship between them. For missing attributes, we use the term "unknown" to complete the sentence. The reconstructed textual descriptions for P05117 are shown below.

---

**Reconstructed Textual Descriptions of P05117**

- *The name of protein is* Polygalacturonase-2 (PG) (PG-2A) (PG-2B) (Pectinase).
- *The organism is* Solanum lycopersicum (Tomato) (Lycopersicon esculentum).
- *The taxonomic lineage of this protein includes* cellular organisms (no rank), Eukaryota (superkingdom), Viridiplantae (kingdom), Streptophyta (phylum), Streptophytina (subphylum), Embryophyta (no rank), Tracheophyta (no rank), Euphyllophyta (no rank), Spermatophyta (no rank), Magnoliopsida (class), Mesangiospermae (no rank), eudicotyledons (no rank), Gunneridae (no rank), Pentapetalae (no rank), asterids (no rank), lamiids (no rank), Solanales (order), Solanaceae (family), Solanoideae (subfamily), Solaneae (tribe), Solanum (genus), Solanum subgen. Lycopersicon (subgenus).
- *The function of this protein includes:* catalytic subunit of the polygalacturonase isozyme 1 and 2

---

(PG1 and PG2); acts in concert with the pectinesterase, in the ripening process; is involved in cell wall metabolism, specifically in polyuronide degradation; the depolymerization and solubilization of cell wall polyuronides mediated by PG2 during ripening seems to be limited by the beta subunit GP1, probably by recruiting PG2 to form PG1.
- *Caution includes* **unknown**.
- *Some miscellaneous things of this protein includes* to avoid liquid rheology of tomato juice, temperature and pressure can be increased to inactivate selectively PG2 during the process.
- *Subunit structure of this protein is* monomer PG2 (isoenzymes PG2A and PG2B); also forms heterodimers called polygalacturonase 1 (PG1) with the beta subunit GP1.
- *The induction of this protein includes* by ethylene.
- *The tissue specificity of this protein is* expressed only in ripening fruits (at protein level).
- *The developmental stage of this protein is* PG1 appears when fruits start to be coloured. When fruits are orange, both PG2 and PG1 are present. In fully ripe fruit, mostly PG2 is expressed.
- *The allergenic properties includes* **unknown**.
- *The biotechnological use includes* the effect of PG can be neutralized by introducing an antisense PG gene by genetic manipulation. The Flavr Savr tomato produced by Calgene (Monsanto) in such a manner has a longer shelf life due to delayed ripening.
- *The pharmaceutical use includes* **unknown**.
- *The diseases it could lead involves include* **unknown**.
- *This protein is usually located in subcellular* secreted, extracellular space, apoplast. Secreted, cell wall.
- *The post-translational modification of this protein includes* N-glycosylated. PG2B isozyme has a greater degree of glycosylation than PG2A.
- *The sequence similarities of this protein includes* belongs to the glycosyl hydrolase 28 family.

## A.2    Dataset preparation

In this section, we detail the construction of the dataset utilized in our experiments. We obtain sequences and their corresponding textual descriptions from Swiss-Prot in UniProt, with data up to March 11th, 2024, under the CC BY 4.0 License. To minimize the similarity between the test sequences and the training sequences, we follow the methodology described in CLEAN [65]. We cluster the data using MMseqs2 [35], applying different sequence identity thresholds to achieve this partition. Here are the steps we follow:

```
# Steq 1. Convert fasta file into mmseqs2 database file
mmseqs createdb sequences.fasta mm_db
# Step 2. Cluster sequences with identity threshold at 10%, 30%, 50%, 70% and 90%,
#         where the threshold variable ${th} is 0.1, 0.3, 0.5, 0.7 or 0.9
mmseqs cluster mm_db mm_clusters results --min-seq-id ${th} -c ${th} --cov-mode 1
# Step 3. Extract clustered sequences to tsv file
mmseqs createtsv mm_db mm_db mm_clusters clusters.tsv
```

This methodology ensures a robust division between training set and test set by employing sequence identity thresholds to control the degree of similarity within clusters. After clustering the sequences, we develop a random-based method for splitting the dataset in an $8 : 1 : 1$ ratio (training:validation:test). This *cluster-guarantee* approach begins by randomly dividing the cluster centers in an $8 : 1 : 1$ ratio. Subsequently, we propose two strategies on each cluster to derive the final dataset: the *k-selected* strategy and the *random-ignore* strategy.

With the *k-selected* strategy, for each key-value cluster (where "key" represents the center sequence of the cluster, and "value" includes all sequences within this cluster), we randomly select $min\{k, len(values)\}$ sequences as the final samples. For the *random-ignore* strategy, we incorporate all sequences from each cluster. Both of these two strategies ensure that the final dataset splits are based on the original clustering, with the validation set and test set equally sharing the remaining clusters.

Practically, we implemented the *k-selected* strategy (where $k$ is set to 1) to minimize redundancy and maximize the diversity of our dataset as much as possible. The sample sizes of the final dataset for each clustering threshold are detailed in Table 9.

Table 9: Statistics of samples in dataset for each clustering threshold.

| Dataset | 10% | 30% | 50% | 70% | 90% |
|---|---|---|---|---|---|
| Training | 5568 | 6412 | 14552 | 30208 | 48988 |
| Validation | 696 | 802 | 1819 | 3776 | 6123 |
| Test | 696 | 802 | 1821 | 3776 | 6124 |

## B  More experiment details

### B.1  Evaluation metrics

In our experiments, we use three important token-level and two region-level evaluation metrics to measure the prediction performance. Three token-level metrics are $\mathbf{F_{max}}$, **AUPRC** and **MCC**. Two region-level metrics are **OS** (Overlap Score) and **FPR** (False Positive Rate).

(1) Following [21], $\mathbf{F_{max}}$ is defined by first calculating the precision and recall for each protein and then taking the average score. For a specific protein $i$, the precision and recall are computed as:

$$\text{precision}_i(\theta) = \frac{|P_i(\theta) \cap T_i|}{|P_i(\theta)|}, \tag{9}$$

and

$$\text{recall}_i(\theta) = \frac{|P_i(\theta) \cap T_i|}{|T_i|}, \tag{10}$$

where $\theta$ is a hyperparameter to adjust the decision threshold, $T_i$ is the set of ground truth active sites for protein $i$, $P_i(\theta)$ is the set of predicted active sites by our model for protein $i$, and $|\cdot|$ denotes the size of the set. Then, the average precision and recall in a protein at threshold $\theta$ are defined as:

$$\text{precision}(\theta) = \frac{1}{M(\theta)} \sum_{i=1}^{M(\theta)} \text{precision}_i(\theta), \tag{11}$$

and

$$\text{recall}(\theta) = \frac{1}{N} \sum_{i=1}^{N} \text{recall}_i(\theta), \tag{12}$$

where $N$ denotes the number of proteins, and $M(\theta)$ denotes the number of proteins on which at least one is made above threshold $t$, *i.e.*, $|P_i(\theta)| > 0$.

Combining these two measures, the maximum F-score is defined as the maximum value of the F-measure over all thresholds. That is,

$$\text{F}_{\text{max}} = \max_{\theta} \left\{ \frac{2 \cdot \text{precision}(\theta) \cdot \text{recall}(\theta)}{\text{precision}(\theta) + \text{recall}(\theta)} \right\}. \tag{13}$$

(2) **AUPRC** (Area Under the Precision-Recall Curve) is a metric representing the pair-centric area under the precision-recall curve. It calculates the average precision scores for all protein-label pairs, which is exactly equivalent to the micro-average precision score for multiple binary classification problems.

(3) The **MCC** (Matthew's Correlation Coefficient) is a metric used to evaluate the performance of binary classification models. It takes into account true positives, true negatives, false positives, and false negatives to assess the model's overall performance. The formula for MCC is:

$$\text{MCC} = \frac{TP \cdot TN - FP \cdot FN}{\sqrt{(TP + FP)(TP + FN)(TN + FP)(TN + FN)}}. \tag{14}$$

Here's what each term represents: $TP$: True Positives; $TN$: True Negatives; $FP$: False Positives; $FN$: False Negatives.

MCC ranges from $-1$ to $+1$, where $+1$ indicates perfect prediction, 0 indicates random prediction, and $-1$ indicates complete disagreement between prediction and observation. It's considered a

balanced measure even when classes are imbalanced, making it a useful metric for evaluating classification models.

(4) Following [8], **OS** (Overlap Score) is defined considering the active sites as a set of non-overlapping segments in a sequence. If $S$ with $|S| = n$ is a sequence of amino acid residues, the active region $A_s$ of $S$ is defined as $A_s = \{(a_i, b_i)\}_i^m$, where $a_i$ and $b_i$ are the index boundaries of the segment $i$. The overlap score between the predicted active region $A = \{(a_{pi}, b_{pi})\}_i^n$ and the ground-truth $A_s = \{(a_{si}, b_{si})\}_i^m$ is defined as:

$$\text{OS} = \frac{\sum_i^n \sum_j^m \max\left(0, \min\left(b_{pi}, b_{sj}\right) - \max\left(a_{pi}, a_{sj}\right)\right)}{\sum_i^m \left(b_{si} - a_{si}\right)}. \tag{15}$$

(5) According to the definition in the last equation, the formula of **FPR** (False Positive Rate) is as follows:

$$\text{FPR} = \frac{\sum_i^n \left(b_{pi} - a_{pi}\right) \mathbb{1}_{\bigwedge_{j=1}^m [a_{pi}, b_{pi}] \cap [a_{sj}, b_{sj}] = \emptyset}}{\sum_i^n \left(b_{pi} - a_{pi}\right)}. \tag{16}$$

## B.2 Truncated length of reconstructed textual descriptions

During the process of feeding $M$ reconstructed textual descriptions into the BLM $f_\psi$, we truncate the lengths of each attributes to optimize the computational resources. This is necessary because some attributes, such as `Function` and `Involvement in Disease`, typically contain more words, whereas others, like `Protein Name` and `Organism`, are usually shorter. Table 10 displays the mean length, 85% length, and truncated length of each attribute in our experiments.

Table 10: Length of each textual description attribute.

| Attribute | Length$_{\text{mean}}$ | Length$_{85\%}$ | Length$_{\text{trunc}}$ |
|---|---|---|---|
| Protein Name | 39.05 | 56 | 48 |
| Organism | 16.22 | 24 | 24 |
| Taxonomic Lineage | 139.82 | 233 | 128 |
| Function | 121.30 | 200 | 128 |
| Caution | 66.99 | 115 | 72 |
| Miscellaneous | 55.80 | 84 | 64 |
| Subunit Structure | 70.94 | 107 | 72 |
| Induction | 45.80 | 69 | 64 |
| Tissue Specificity | 48.92 | 76 | 64 |
| Developmental Stage | 59.85 | 93 | 64 |
| Allergenic Properties | 38.89 | 74 | 48 |
| Biotechnological Use | 92.91 | 155 | 128 |
| Pharmaceutical Use | 55.67 | 68 | 64 |
| Involvement in Disease | 212.02 | 330 | 256 |
| Subcellular Location | 47.62 | 75 | 64 |
| Post-translational Modification | 74.63 | 124 | 96 |
| Sequence Similarities | 27.45 | 35 | 32 |

## B.3 Introduction of baseline models

**ESM-1b** is a deep learning model based on the Transformer architecture, designed specifically for processing and understanding protein sequences. It is trained on an extensive dataset comprising 250 million protein sequences, totaling around 86 billion amino acids, leveraging unsupervised learning techniques. The code and weight can be accessed from `https://github.com/facebookresearch/esm` under the MIT License.

**ESM-1v** is a language model specialized for predicting the functional effects of sequence variations, enabling state-of-the-art zero-shot predictions. It shares the same architecture as ESM-1b but is specifically trained on the UniRef90 dataset to enhance its predictive capabilities for variant effects. The code and weight can be accessed from `https://github.com/facebookresearch/esm` under the MIT License.

**ESM-2** is trained on protein sequences from UniRef database. It is capable of predicting structure, function, and various other properties of proteins directly from individual sequences. In our experiments, we use the 650M parameter version of the ESM-2. The code and weight can be accessed from `https://github.com/facebookresearch/esm` under the MIT License.

**ProtT5** is a model developed by training the T5 [43] architecture from NLP on protein sequences. There are two versions of this model, each trained on BFD100 and UniRef50 respectively. Additionally, there are two size variants of the model, and we utilize the XL size version. The code and weight can be accessed from `https://github.com/agemagician/ProtTrans` under the Academic Free License v3.0.

**ProtBert** is a model that has been trained on protein sequences using the BERT [10] architecture from NLP. Similar to ProtT5, there are also two versions of ProtBert, each trained on BFD100 and UniRef 100 respectively. The code and weight can be accessed from `https://github.com/agemagician/ProtTrans` under the Academic Free License v3.0.

**ProtAlbert** is another model trained on protein database, UniRef100, using the Albert language model architecture [26] in NLP. Albert reduces BERT's complexity by hard parameter sharing between its attention layers which allows to increase the number of attention heads. The code and weight can be accessed from `https://github.com/agemagician/ProtTrans` under the Academic Free License v3.0.

**ProtXLNet** is trained on UniRef100 database following the successful language model architecture XLNet [63] in NLP. The code and weight can be accessed from `https://github.com/agemagician/ProtTrans` under the Academic Free License v3.0.

**ProtElectra** is also trained on UniRef100 database but follows the Electra [4] architecture. Electra tries to improve the sampling-efficiency of the pretraining task by training two networks, a generator and a discriminator. The generator reconstructs masked tokens, potentially creating plausible alternatives, and the discriminator detects which tokens were masked. In our experiments, we use the discriminator of ProtElectra to extract residue-level features. The code and weight can be accessed from `https://github.com/agemagician/ProtTrans` under the Academic Free License v3.0.

**PETA** is a benchmark designed to evaluate the impact of vocabulary size and tokenization methods on the transfer learning capabilities of protein language models. The benchmark facilitates systematic assessments of how different training configurations of protein language models affect their performance in biologically relevant downstream applications. In our experiments, we utilize the amino acid-level version, `deep_base`, as the baseline for comparison. The code and weight can be accessed from `https://github.com/ginnm/ProteinPretraining` under the MIT License.

**S-PLM** is a structure-aware protein language model that efficiently leverages both sequence and structural data during its training phase. This integration is achieved through multi-view contrastive learning, which enables the model to deeply understand the complex interplay between a protein's sequence and its structure. During inference stage, S-PLM requires only the sequence data for inference, eliminating the need for structural input. The code and weight can be accessed from `https://github.com/duolinwang/S-PLM` under the MIT License.

**TAPE** is a benchmarking framework designed to systematically evaluate the performance of semi-supervised learning models on protein sequences. It comprises five biologically relevant tasks that span different domains of protein biology, focusing on the generalization capabilities of protein embeddings. The code and weight can be accessed from `https://github.com/songlab-cal/tape` under the BSD 3-Clause License.

**MIF** is a structured graph neural network that improves protein representation learning by utilizing protein backbone structures during pretraining. It reconstructs masked protein sequences with the help of structural information, enhancing its ability to capture complex biological properties. MIF-ST extends MIF by incorporating sequence data from a pretrained sequence-only protein language model. This model leverages both structural and sequence information, enhancing training effectiveness and broadening its application. The code and weight can be accessed from `https://github.com/microsoft/protein-sequence-models` the under 1-clause BSD License.

**PST** is a refined model that enhances transformer-based protein language models by integrating structural information through structure extractor modules within its self-attention architecture. It is pretrained on protein structure databases, using a traditional masked language modeling objective. It

has two versions of model, "Standard" and "Train struct only", with a number of size. We utilize the version of `pst_t33` and `pst_t33_so` in our experiments. The code and weight can be accessed from `https://github.com/BorgwardtLab/PST` under the BSD 3-Clause License.

**ProtST** is a framework designed to enhance the pretraining and understanding of protein sequence representations by integrating multi-modal learning with biomedical texts. It employs the Prot-Describe dataset, which pairs protein sequences with four textual descriptions of their properties derived from the Swiss-Prot database. We use the ProtST-ESM-1b version and ProtST-ESM-2 version in our experiments. The code and weight can be accessed from `https://github.com/DeepGraphLearning/ProtST` under the Apache-2.0 License.

### B.4 Examples of generated descriptions by agent model

In practical scenarios, when encountering newly discovered protein, one common challenge is the absence of high-quality textual annotations. To address this issue, we utilize state-of-the-art biomedical text generation method, Prot2Text[5], which serves as an agent model, to generate corresponding descriptions of proteins, as illustrated in Figure 6 to complement missing text modality. These generated descriptions have encapsulated key information about protein properties, including functions, locations and so on. By integrating these generated texts into our models, we significantly enhance the capability of the PLMs in active sites identification, leading to improved performance and deeper insights into protein.

---

**Sequence:** NLYQFGKMINHMVGKSPIFSYGDYGCYCGWGGKGTPVDATDRCCFVHDCCYGRANGCDPKLSTYSYNFQNGNIVCGNKYGCLRHI CECDRVAAICFGENVNTYDKKFLSSSRCRQTSEQC

**Agent:** Basic phospholipase A2 that inhibits ADP-, thrombin- and arachidonic acid-induced platelet aggregation. Inhibits phospholipase A2 activity in the presence of collagen, and inhibits the binding of monolayers inducing aggregation induced by collagen. Inhibits the formation of superoxide anions in vivo and in vitro. Inhibits the adhesion of endothelial cells to fibronectin and collagen. Inhibits cell migration and invasion in vitro. Inhibits the adhesion of endothelial cells to collagen and human umbilical vein endothelial cells in vitro. Inhibits the adhesion of monocytes to collagen in vitro. Inhibits the binding of human platelets to fibrinogen. Inhibits the adhesion of human platelet receptors to fibrinogen in vitro. Inhibits the binding of human monocytes to dendritic cells in a dose-dependent manner. Inhibits the binding of human monocytes to dendritic cells in a dose-dependent manner. Inhibits the binding of human plasma fibronectin to TNFRSF10 in a dose-dependent manner …

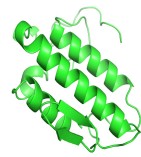

---

**Sequence:** MADEEKLPPGWEKRMSRSSDRVYYFNHITNASQWERPSGNSSSGGKNGQGEPARVRCSHLLVKHSQSRRPSSWRQEKITRTKEEALE LINGYIQKIKSGEEDFESLASQFSDCSSAKARGDLGAFSRGQMQKPFEDASFALRTGEMSGPVFTDSGIHIILRTE

**Agent:** Peptidyl-prolyl cis/trans isomerase (PPIase) that binds to and isomerizes specific phosphorylated Ser/Thr-Pro (pSer/Thr-Pro) motifs. By inducing conformational changes in a subset of phosphorylated proteins, acts as a molecular switch in multiple cellular processes. By inducing conformational changes in a subset of NEDD8-dependent transcription factors, acts as a molecular switch to regulate transcription in neurons. Inhibits the activity of several kinases, including MAPKs, IRF3 and MAPK1/ERK2 and MAPK3/ERK1. Acts as a regulator of JNK activation by binding to phosphorylated JNK activation factor (JUN) and inhibiting its activity. Inhibits the ubiquitin ligase activity of the type I interferon regulatory factor IRF3 and promotes its ubiquitination and subsequent degradation. Inhibits the activity of the phosphorylation-dependent protein kinase activity of the heat shock protein RELA. Inhibits the activity of STMN1/p38 and STMN2/p65 ATPase activity by binding to phosphorylated L-proline residues …

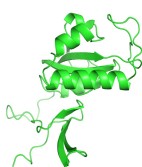

---

**Sequence:** MAAAVLLAVGLRAARRTLAAAGARGAQVRGNAGVSDGSEVAKAQKAAPGGASPTIFSRILDRSLPADILYEDQQCLVFRDVAPQAP VHFLVIPRKPIPRISQAEEDDQQLLGHLLLVAKKIAQAQGLKDGYRLVVNDGKMGAQSVYHLHIHVLGGRQLQWPPG

**Agent:** Exhibits adenosine 5'-monophosphoramidase activity, hydrolyzing purine nucleotide phosphoramidates with a single phosphate group such as adenosine 5'monophosphoramidate (AMP-NH2), adenosine 5'monophosphomorpholidate (AMP-morpholidate), purine nucleotide phosphoramidate (NMP-morpholidate) and adenosine 5'monophosphomorpholidate (AMP-cAMP). Hydrolyzes purine nucleoside tryptamine phosphoramidates with a single phosphate group such as inositol 1,4-bisphosphate (Ins(1,4)P2) and 3',5'-cyclic AMP-morpholidate (3',5'-cAMP). Hydrolyzes lysyl endopeptidase (LENAT) substrates such as 2,3-diamino-2-pyridine carboxylic acid methyl ester phosphoramidate (2,3-diamino-2-pyridine fluoromethyl ester phosphoramidate) and p-nitrophenylphosphoramidate methyl ester phosphoramidate (pNPP-morpholidate). Hydrolyzes fluorogenic purine nucleoside tryptamine phosphoramidates methylated at Lys-Arg-Ala-Lys (Ala-Lys-Ala-Arg-AMP) and arginine phosphoramidate …

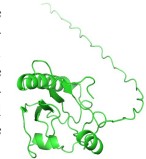

---

**Sequence:** MAAAMDVDTPSGTNSGAGKKRFEVKKWNAVALWAWDIVVDNCAICRNHIMDLCIECQANQASATSEECTVAWGVCNHAFHFHCI SRWLKTRQVCPLDNREWEFQKYGH

**Agent:** E3 ubiquitin ligase component of multiple cullin-RING-based E3 ubiquitin-protein ligase (CRLs) complexes which mediate the ubiquitination and subsequent proteasomal degradation of target proteins, including proteins involved in cell cycle progression, signal transduction, transcription and transcription-coupled nucleotide excision repair. E3 ubiquitin ligase complexes accept ubiquitin from an E2 ubiquitin-conjugating enzyme in the form of a thioester and then directly transfers the ubiquitin to targeted substrates. It probably triggers the ubiquitin-mediated degradation of different substrates. In the cytoplasm, the functional specificity of the E3 ubiquitin-conjugating enzyme depends on the variable substrate recognition components. It is for instance, required for the formation of the CRLs complexes from the CDC34-CDC57 ubiquitinated histone H2B. The functional specificity of the E3 ubiquitin-conjugating enzyme depends on the variable substrate recognition components …

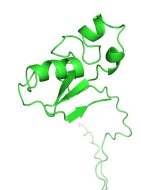

---

Figure 6: Examples of generated descriptions by the agent model.

---

[5]Code and weight can be accessed from `https://github.com/hadi-abdine/Prot2Text` under the CC BY-NC-SA 4.0 License.

## C Explanation for the biological principle

In biology, "*Similar protein sequences give rise to similar structures and functions*" means that if two proteins have similar sequences of building blocks (amino acids), they will likely have similar shapes and do similar jobs in the body. For example, consider two proteins **hemoglobin** and **myoglobin**. Both proteins have similar amino acid sequences in their oxygen-binding regions, which means they fold into similar shapes. Because of this similarity, both proteins are able to bind oxygen, but they do it in different ways suited to their specific roles: hemoglobin transports oxygen in the blood, while myoglobin stores oxygen in muscles. This illustrates how similar sequences lead to similar structures and functions.

## D More experiment results

### D.1 Performance comparison with other methods

In order to evaluate the performance of our approach more convincingly, we also select other new PLM (*i.e.*, SaProt [49]) as well as other classical methods (Random Forest and Support Vector Machine) [64] and statistics-based approach (DISCERN [46]) for comparison. The results of the comparison are shown as Table 11.

Table 11: Performance comparison with other methods.

| Method | $F_{max}\uparrow$ | AUPRC $\uparrow$ | MCC $\uparrow$ | OS $\uparrow$ | FPR $\downarrow$ |
|---|---|---|---|---|---|
| **MMSite** | **0.8250** | **0.8909** | **0.8319** | **0.8549** | **0.1689** |
| SaProt (650M-AF2) | 0.7181 | 0.8562 | 0.7259 | 0.7480 | 0.2731 |
| Random Forest | 0.4339 | 0.5545 | 0.4382 | 0.3137 | 0.2825 |
| Support Vecotr Machine | 0.4270 | 0.4708 | 0.4017 | 0.3409 | 0.4000 |
| DISCERN | 0.2870 | 0.4539 | 0.2949 | 0.1921 | 0.4176 |

### D.2 The quality of texts generated by Prot2Text

To evaluate the performance of Prot2Text, we built a dataset of 1k proteins randomly sampled from ProTAD, as shown in Table 12. The performance on ProTAD-1k is good and generally consistent with the reported performance, indicating that Prot2Text can readily serve as an agent to generate textual descriptions for proteins.

Table 12: The quality of texts generated by Prot2Text.

| | BLEU Score | Rouge-1 | Rouge-2 | Rouge-L | BERT Score |
|---|---|---|---|---|---|
| Prot2Text (reported) | **36.51** | 53.68 | 45.60 | 51.40 | 85.20 |
| Prot2Text on ProTAD-1k | 25.73 | **56.56** | **49.46** | **54.15** | **86.57** |

Table 13 below compares the performance of MMSite using human-annotated and Prot2Text-generated texts for training. The performance of two settings was similar across most metrics.

Table 13: Performance of MMSite using two types of text for training.

| Method | $F_{max}\uparrow$ | AUPRC $\uparrow$ | MCC $\uparrow$ | OS $\uparrow$ | FPR $\downarrow$ |
|---|---|---|---|---|---|
| ProTAD (human annotation) | **0.8250** | 0.8909 | **0.8319** | **0.8549** | **0.1689** |
| Prot2Text generation | 0.8194 | **0.8923** | 0.8254 | 0.8433 | 0.1751 |

### D.3 Impact of the length of protein sequence

Here we examine the impact of varying the maximum input sequence length on the performance of MMSite. We **retrain** our model using different sequence input length, specifically 128, 256, 512, 768, and 1024 amino acids, with the clustering threshold of 10%, as illustrated in Figure 7. When the input length is 768 or 1024, the batch size is set to 16, and for other input lengths, it is 24. In this ablation study, samples will be removed if there is no active site within the specified length range.

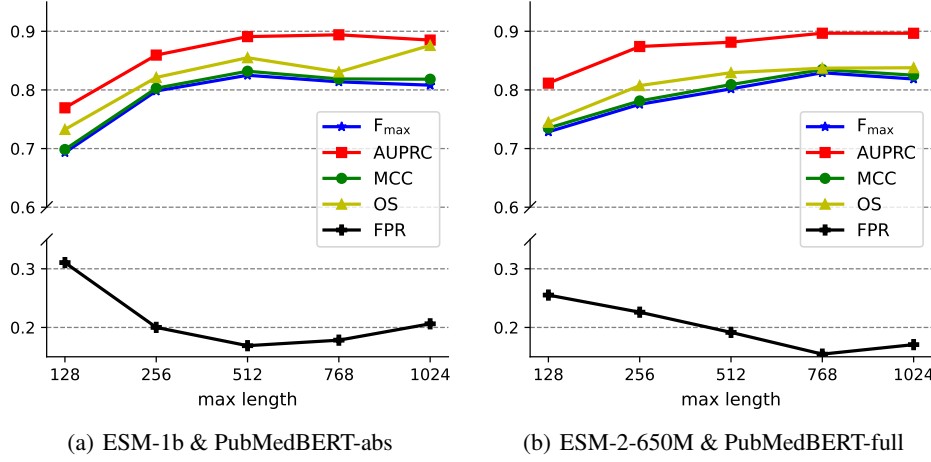

(a) ESM-1b & PubMedBERT-abs   (b) ESM-2-650M & PubMedBERT-full

Figure 7: Performance on different maximum input sequence length.

It is apparent that the model's performance tends to be quite good when the maximum length is set to 512 or higher. However, when the maximum length is set below 256, the model's performance declines due to the limited number of samples—in fact, there are only 786 samples in the training dataset. Despite this decrease, the performance remains at a high level, demonstrating MMSite's robustness under varying training input conditions.

## D.4 Quantitative results

As complements to Table 1, we report comparisons of MMSite in dataset where clustering threshold is 30% and 50% with other 14 baselines in Table 14 and 15, which perform best in 10% case. It can be seen that MMSite still best in the 30% case. Although MMSite does not lead on all metrics at a 50% threshold, its performance is still noteworthy. It is important to notice that increasing the threshold significantly raises the risk of data leakage. Specifically, the dataset size at a 50% threshold is nearly three times larger than that at a 10% threshold (as shown in Table 9). We think their performances are closer at 50% threshold because: (1) The dataset is easy enough for the baseline models to generalize; (2) The performance is already very high, leaving little room for improvement. Therefore it is more reasonable to draw comparisons at lower thresholds. From this perspective, MMSite excels in conditions with limited data, demonstrating the benefits of leveraging textual modalities. This showcases MMSite's effectiveness and robustness in handling inadequate data scenarios.

Table 14: Comparison on the dataset with clustering threshold at 30%.

| Input | Method | Version | $F_{max}$ ↑ | AUPRC ↑ | MCC ↑ | OS ↑ | FPR ↓ |
|---|---|---|---|---|---|---|---|
| Seq. | ESM | 1b | 0.7679 | 0.8769 | 0.7749 | 0.7776 | 0.2094 |
| | | 1v | 0.7146 | 0.8481 | 0.7222 | 0.7349 | 0.2668 |
| | | 2-650M | 0.7476 | 0.8630 | 0.7537 | 0.7684 | 0.2441 |
| | ProtAlbert | | 0.6219 | 0.7630 | 0.6302 | 0.6339 | 0.3454 |
| | PETA | deep_base | 0.6986 | 0.8186 | 0.7041 | 0.6986 | 0.2753 |
| | S-PLM | | 0.7700 | 0.8990 | 0.7757 | 0.7694 | 0.2033 |
| Seq. & Struct. | MIF | MIF | 0.1143 | 0.3609 | 0.1465 | 0.1385 | 0.8399 |
| | | MIF-ST | 0.1067 | 0.2888 | 0.1068 | 0.1064 | 0.8925 |
| | PST | t33 | 0.7153 | 0.8607 | 0.7221 | 0.7159 | 0.2514 |
| | | t33_so | 0.7430 | 0.8648 | 0.7498 | 0.7634 | 0.2420 |
| Seq. & Text | ProtST w/ retrain | ESM-1b | 0.5006 | 0.7299 | 0.5105 | 0.5999 | 0.4548 |
| | | ESM-2 | 0.5356 | 0.7418 | 0.5505 | 0.5999 | 0.4548 |
| | ProtST w/o retrain | ESM-1b | 0.5043 | 0.7497 | 0.5125 | 0.5206 | 0.4700 |
| | | ESM-2 | 0.6510 | 0.8217 | 0.6592 | 0.6633 | 0.3202 |
| | **MMSite** | | **0.8470** | **0.9001** | **0.8525** | **0.8702** | **0.1483** |

Table 15: Comparison on the dataset with clustering threshold at 50%.

| Input | Method | Version | $F_{max}$ ↑ | AUPRC ↑ | MCC ↑ | OS ↑ | FPR ↓ |
|---|---|---|---|---|---|---|---|
| | | 1b | 0.9074 | 0.9565 | 0.9112 | 0.9234 | 0.0905 |
| | ESM | 1v | 0.8899 | 0.9373 | 0.8928 | 0.8987 | 0.1034 |
| | | 2-650M | **0.9198** | 0.9577 | **0.9223** | 0.9258 | **0.0735** |
| Seq. | ProtAlbert | | 0.8050 | 0.8827 | 0.8090 | 0.8191 | 0.1826 |
| | PETA | deep_base | 0.8601 | 0.9080 | 0.8642 | 0.8739 | 0.1340 |
| | S-PLM | | 0.9123 | 0.9585 | 0.9155 | 0.9232 | 0.0827 |
| Seq. & Struct. | MIF | MIF | 0.5180 | 0.6832 | 0.5272 | 0.4974 | 0.4204 |
| | | MIF-ST | 0.4214 | 0.6095 | 0.4290 | 0.4017 | 0.5236 |
| | PST | t33 | 0.9102 | **0.9608** | 0.9125 | 0.9126 | 0.0813 |
| | | t33_so | 0.9129 | 0.9589 | 0.9155 | 0.9204 | 0.0823 |
| Seq. & Text | ProtST w/ retrain | ESM-1b | 0.3909 | 0.6818 | 0.4009 | 0.4034 | 0.5712 |
| | | ESM-2 | 0.2639 | 0.4604 | 0.2690 | 0.2655 | 0.7126 |
| | ProtST w/o retrain | ESM-1b | 0.5300 | 0.7284 | 0.5388 | 0.5386 | 0.4351 |
| | | ESM-2 | 0.5719 | 0.7729 | 0.5795 | 0.5850 | 0.4013 |
| | MMSite | | 0.9190 | 0.9538 | 0.9220 | **0.9343** | 0.0812 |

## D.5 Inference performance comparison

During inference, if the protein has corresponding multi-attribute text descriptions, the process is the same as during training. However, if such descriptions are not available, we use the Prot2Text agent model to generate the text inputs. Table 16 is a comparison of MMSite inference time between using pre-existing text descriptions from ProTAD and generated text with an agent model. For further context, we also compare against the BioMedGPT [32], another excellent protein-to-text model. The tests were conducted on a single GPU (NVIDIA GeForce RTX 4090) and a CPU (Intel(R) Xeon(R) Platinum 8375C CPU @ 2.90GHz).

Table 16: Inference time comparison for different text sources.

| Source of Text | Average GPU time (s/sample) | Average CPU time (s/sample) |
|---|---|---|
| ProTAD | **0.1336** | **1.7523** |
| Prot2Text generation | 0.9044 | 4.2963 |
| BioMedGPT generation | 5.1136 | 71.169 |

We also compared the model performance using text generated by Prot2Text and BioMedGPT as Table 17.

Table 17: Performance comparison between Prot2Text and BioMedGPT generated text.

| Source of Text | $F_{max}$ ↑ | AUPRC ↑ | MCC ↑ | OS ↑ | FPR ↓ |
|---|---|---|---|---|---|
| Prot2Text generation | **0.8250** | 0.8909 | **0.8319** | **0.8549** | **0.1689** |
| BioMedGPT generation | 0.8230 | **0.8921** | 0.8304 | 0.8540 | 0.1693 |

As the comparison shows, while using Prot2Text does increases the inference time of ProTAD, it provides comparable performance to BioMedGPT with a much smaller inference time cost.

# E  Limitations & Broader impacts

In our research, we utilize the data from Swiss-Prot in UniProt, which is renowned for its high-quality and expertly curated annotations, ensuring a high level of confidence in the dataset. Our study is able to effectively identify protein active sites, yet it does not specify the biochemical reactions for each, due to the inherent scarcity of detailed annotations from challenging biological wet lab

experiments. Nonetheless, our approach, which integrates multi-modal deep learning, encourages further exploration into protein reaction mechanisms, setting a foundation for more targeted and comprehensive future research. However, it is worth noting that our powerful pretrained model may potentially be misused for harmful purposes, such as the design of dangerous drugs. We anticipate that future studies will address and mitigate these concerns.

# F  More visualisation results

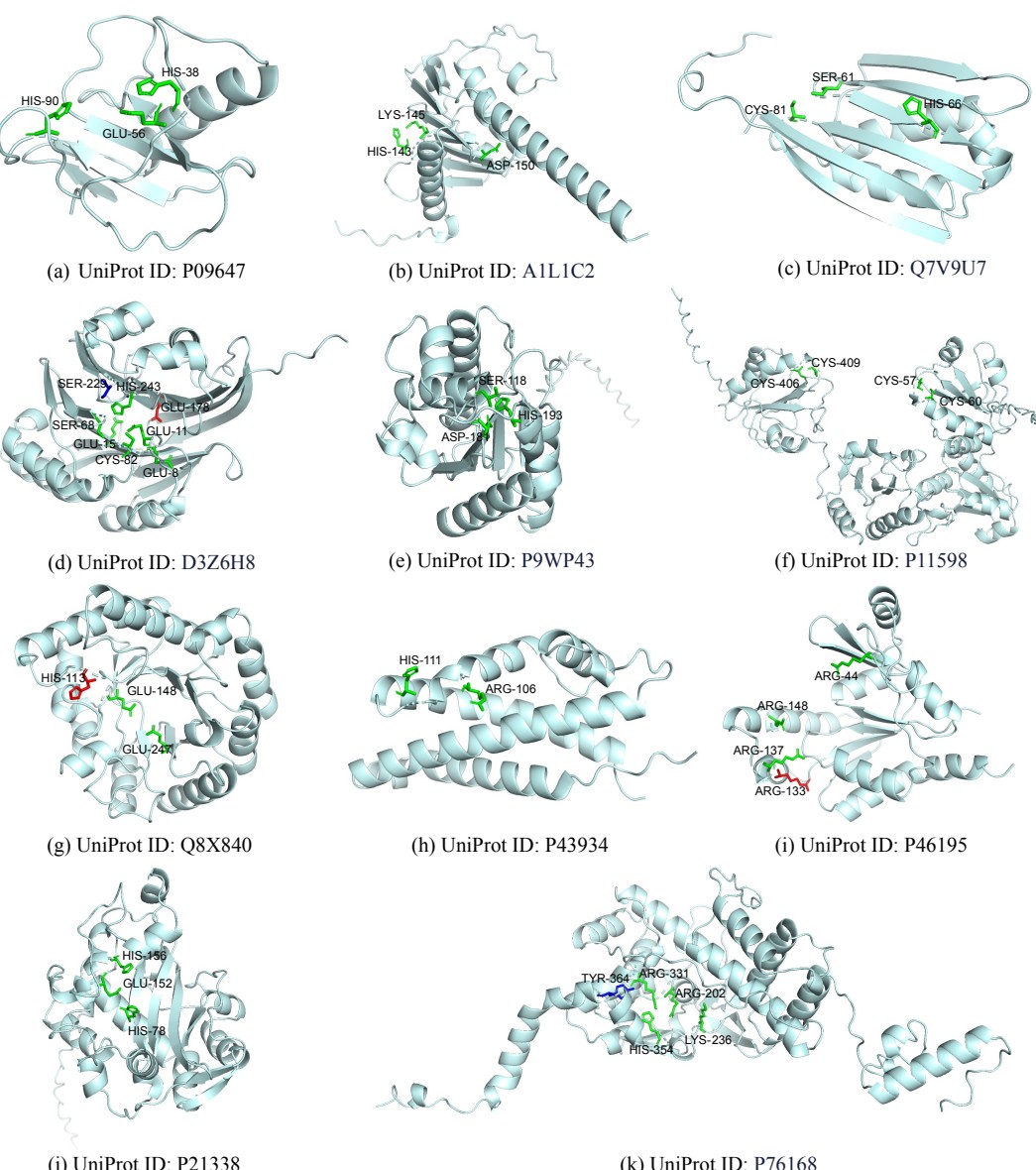

Figure 8: Visualisation of active site identification results of proteins by MMSite. Each subfigure caption is the protein's Entry ID in the UniProt database. The colors mean the same as in Figure 3.

