# OpenReview forum: "MMSite: A Multi-modal Framework for the Identification of Active Sites in Proteins"
_NeurIPS.cc/2024/Conference — NeurIPS 2024 poster_

### Official Review · Reviewer_sxh3 · 2024-07-01

**Soundness:** 2
**Presentation:** 2
**Contribution:** 3
**Rating:** 5
**Confidence:** 4

**Summary:**

The manuscript proposes MMSite, a multimodal framework combining amino acid sequences and textual descriptions to predict active sites of proteins at a residue-level. For that, the authors train an attention-based architecture in two stages. In the first stage, the objective is to align sequence and text embeddings. While in the second stage, it fuses both sequence and text embeddings to predict the final active sites. To train the model, the authors curated a dataset named ProTAD with sequence, text, and active site date. Results show that the proposed MMSite framework improves the performance for active site prediction over sequence-based and sequence-structure-based baselines.

**Strengths:**

The main strengths of the paper are as follows:

1. The authors curated a dataset named ProTAD, in which protein sequences, their respective text annotations, and residue-level active site annotations are made available.
2. The authors introduce a two-stage multimodal framework that aligns sequence and textual embeddings, and then fuse these embeddings for active site predictions.
3. Results obtained by authors show that even using a predicted textual context at inference time, the proposed method can enhance the performance for active site prediction over baseline methods with prediction heads trained for the same task. Extensive ablation studies were performed for the proposed methodology.

**Weaknesses:**

The main weaknesses of the paper are as follows:

1. As text descriptions are not available during the inference stage for novel proteins, the proposed methodology relies on an external model to generate the textual context for inference.
2. Information in the manuscript seems to lack sufficient details to replicate the implementation of the proposed architecture in MMSite and training the baseline methods for the active site prediction task.
3. The authors show that their method achieves the best results when splitting the datasets using an identity threshold equal to 10%. When a higher threshold is used, the gap between the proposed method and baseline methods is smaller. The paper would be more strong with deeper reasoning and analysis regarding this gap in performance.

**Questions:**

My main questions/comments are as follows:

1. (Related Work Section) Methods that learn embeddings using sequence and structure input are also multimodal. These should also be properly reviewed in the related works section. Also, probably it should highlighted that in the proposed work the multimodality is sequence and text, and accordingly compare the proposed method with the other sequence-text model ProtST used for comparison.

2. (Baseline Comparison) ProtST is also a multimodal framework, but its results are much lower when compared to MMSite. Any reasoning for the large difference in performance?

3. (Lines 129-130, Line 149) I would suggest explaining already in the Introduction that during inference only the protein sequence is the input, and also that the proposed framework uses the embeddings from pre-trained PLMs and BLMs as part of its methodology.

4. (CLS token) I would suggest adding a reference on why only the CLS token is used for part of the attributes in the MACross module.

5. (Neural Network Architecture) It is very hard to understand the dimensions after each module of the architecture, which makes it hard to understand the intuition behind modules such as the MACross and the inter-modality attention. Because of this lack of sufficient information for the reader, it is hard to understand sentences such as the one in Line 179: "Inspired by [19], we adopt cross-modal soft-label alignment to make one-hot label continuous.".

6. (Neural Network Architecture) In Fig. 2 during the fusing step, there is a skip concatenate and prediction module that also takes as input the embeddings from the sequence-based feature extractor. How does this skip, concatenate, and prediction module work? The reviewer is not able to understand this part with the explanation in lines 189-201.

7. (Table 1) For the results in Table 1, how the baseline methods were trained. It is mentioned that a prediction head using a Transformer architecture is used. Is the embeddings for them extracted and then a prediction head trained using the proposed dataset? For the standard deviation how many times each method was trained with different random seeds?

8. (Reproducibility) Is code available in an anonymized repository for reviewers or will be available later upon acceptance?

9. (Prompt Ablation) Is there a prompt ablation or some evidence that the manual prompting is necessary to input the text descriptions to the BLM?

10. (Lines 176-180) It is not clear how the use of the KL divergence of the cosine similarities solves the problem of the possibility of "similar protein sequences giving rise to similar structures and functions" in a given batch. Can you give additional intuition regarding the modeling of this loss function for the alignment phase?

11. (Dataset splits) It is observed in the results that even for clustering thresholds as low as 10% the proposed method generalizes better than other methods, while for higher thresholds performance is similar to ESM. It seems that the text description conditions the model to learn the right site for a given function, acting as an implicit condition in this case based on the performance of the pre-trained BLM. It seems as a weakness for the design of de novo methods when compared to surface methods like MaSIF or sequence-structure methods like ScanNet and PeSTO. How would the proposed method be compared to these methods in terms of binding site interface prediction and generalization to de novo proteins?

Minor Comments:

1. Acronyms: PLMs need to be defined in line 32 of the manuscript.

2. Writing: There are issues in writing. Typos occur, e.g. line 133 "serveral", line 223 "will be removed", line 105 "Sites". There are expressions and words that would be better rewritten in a descriptive manner, e.g. "obvious", "near perfection", "it is clear", "after many attempts". The sentence in lines 114-115 should be rewritten.

3. Notations: The notation for sequence in line 124 could follow the same pattern as the attribute in line 126 and the annotation target in line 127.

4. Figures: Figure 1 does not seem to be mentioned in the text. The blue arrows for inference in Figure 2 are confusing and it is hard to grasp the dimensions obtained after each block. The labels in Figure 3 seem to be confusing and Figure 8 lacks proper captions.

5. References/Format: references should follow the right format, both in text and in the reference section.

**Limitations:**

The authors address potential negative societal impacts and briefly explore the limitations of the proposed methodology in the Appendix. The paper would benefit by providing the source code for reviewers to evaluate the reproducibility of the proposed method.

---

> ### Author Rebuttal · Authors · 2024-08-07
>
> Dear Reviewer,
>
> Thank you for your valuable feedback and constructive suggestions. Below is our point-by-point response to your comments.
>
> **Re - Weaknesses 1**
>
> It's true that MMSite rely on an external generative model to produce textual context for inference. This decouples the training of protein-to-text models from that of the active site prediction model. Training MMSite on top of the pretrained BLM and PLM models is very energy-efficient, so we could easily get a better active site prediction model once a better protein-to-text model is available.
>
> **Re - Weaknesses 2**
>
> We apologize for the confusion.
> In Section 4.1, we have described the implementation of each component of MMSite, and covered the training strategy for baseline methods in Lines 236-238.
>
> **Re - Weaknesses 3**
>
> When the split strategy is harsh, the baseline methods couldn't generalize, whereas MMSite can find meaningful patterns from both sequence and text and achieve good performance. We think their performances are closer at 50% threshold because (1) the dataset is easy enough for the baseline models to generalize, and (2) the performance is already very high, leaving little room for improvement. We will add this discussion to the revised paper.
>
> **Re - Questions 1**
>
> Thank you for your valuable suggestions. We will include a review of sequence-structure multimodal methods in the Related Work section, and emphasize the modalities of MMSite and compare it with similar sequence-text multimodal methods.
>
> **Re - Questions 2**
>
> While ProtST is a protein-text multimodal model, it aligns global sequence-level representations with textual descriptions, overlooking individual amino acid features, which reduces their effectiveness in residue-level tasks. In contrast, MMSite integrates text-empowered sequence representations directly at the token level, preserving the original features of amino acids. This ensures the model can benefit from the information provided by the text descriptions.
>
> **Re - Questions 3**
>
> In Lines 61-62, we have covered the inference strategy, but did not emphasize the source of embeddings, which may cause confusion. We will clarify both points in the future version. Thank you for your suggestion.
>
> **Re - Questions 4**
>
> The [CLS] token is used to aggregate the sequence-level representation of the entire input sentence, as first proposed in BERT. We will add this reference in the future version.
>
> **Re - Questions 5**
>
> We apologize for the confusion. Due to the character limit, we have provided detailed explanations about the dimensions in the **Author Rebuttal**. For Line 179, please refer to our response in the Re - Question 10 section.
>
> **Re - Questions 6**
>
> During the fusing step, we concatenate the text-enhanced sequence features with the original sequence features along the feature dimension. This combined feature is then fed into the prediction module to produce the final predictions. Detailed dimensions are provided in the **Author Rebuttal**. Additionally, for newly discovered proteins that lack text descriptions, we use an agent model to generate the necessary texts, which are then used alongside the sequences.
>
> **Re - Questions 7**
>
> Yes, we first extract amino acid-level embeddings from the baseline models, and then use a Transformer, followed by an MLP and a Sigmoid function to obtain active site probabilities. For the standard deviation, we trained each method 5 times with different random seeds.
>
> **Re - Questions 8**
>
> We have already included well-organized code, dataset and detailed instructions for replication in a zip file as **Supplementary Material**.
>
> **Re - Questions 9**
>
> We have tested the effect of removing manual prompting. The comparison is shown below:
>
> |Method|F_max|AUPRC|MCC|OS|FPR|
> |:-:|:-:|:-:|:-:|:-:|:-:|
> |w manual prompting|**0.8250**|0.8909|**0.8319**|**0.8549**|**0.1689**|
> |w/o manual prompting|0.8157|**0.8911**|0.8221|0.8460|0.1793|
>
> The results show that manual prompting improves performance on most metrics. We believe this is because: (1) complete sentences provide richer context for the BERT-based BLM, (2) it reduces ambiguity in attribute meanings, and (3) it aligns better with BLM's pretraining, leveraging its knowledge more effectively.
>
> **Re - Questions 10**
>
> Traditional hard-label alignment assigns positive pairs a label of 1 and negative pairs a label of 0, pushing them apart in high-dimensional space. This isn't ideal for similar protein sequences $S_i$ and $S_j$, whose corresponding texts $T_i$ and $T_j$ are also similar. Thus, pushing $S_i$ away from $T_j$ may not be reasonable. To address this issue, we align the distribution $Q_i^{\rm s2t}$ towards $P_i^{\rm s2s}$ and $Q_i^{\rm t2s}$ towards $P_i^{\rm t2t}$ using KL divergence in Equation 8. This allows for a more nuanced alignment that respects the inherent similarities within the sequence and textual domains.
>
> **Re - Questions 11**
>
> Surface-based and sequence-structure methods require structural information, which can be costly to obtain. In contrast, functional information may be more accessible for de novo proteins.
>
> Empirically, MMSite performing better at low clustering threshold demonstrates its generalization capabilities, but does not imply it is ineffective to de novo proteins.
> That said, we have not evaluated MMSite on de novo proteins, so we are not 100% sure.
>
>
> **Re - Questions Minor Comments**
>
> We will address all the issues you point out in the revised version to ensure they are correct and clear. Specifically:
> - For Figure 2, please refer to our newly submitted PDF.
> - For Figure 3, we use three colors to show the model's results: green for correctly predicted sites, blue for sites not predicted, and red for incorrectly predicted sites.
> - For Figure 8, we will add more details the caption: "Each subfigure caption is the protein's Entry ID in the UniProt database. The colors mean the same as in Figure 3."
>
> Thank you again for your time and consideration.
>
> Best regards,
>
> Authors

---

> > ### Comment · Reviewer_sxh3 · 2024-08-09
> > **Re: Rebuttal by Authors**
> >
> > Thank you for your detailed responses and additional experiments.
> >
> > I think my concerns and comments are mostly addressed.
> >
> > As additional minor comments:
> >
> > 1. Writing comments addressed by other reviewers should be addressed.
> > 2. I suggest modifying the captions in Fig. 3 and 8 to clarify the meaning of each color.
> > 3. Regarding Question 11, I still feel that the dataset for pre-training the pLM and BLM has an influence on the generalization of MMSite. Does the performance of Prot2Text is one of the bottlenecks for the application of this methodology to de novo proteins?
> >
> > I have increased my score.

---

> > > ### Author Response · Authors · 2024-08-10
> > >
> > > Dear Reviewer,
> > >
> > > We sincerely appreciate your positive feedback and thoughtful suggestions.
> > >
> > > We will make sure to address the writing issues and modify the captions to further improve the clarity and quality of our manuscript.
> > >
> > > Regarding Question 11, we agree that the dataset used for pre-training the PLM and BLM can significantly influence the generalization of MMSite. The performance of Prot2Text indeed plays a crucial role in the application to de novo proteins. While Prot2Text is effective in generating textual descriptions, its performance could affect the accuracy of predictions for proteins without existing annotations. We will continue to explore ways to mitigate this and improve the robustness of our approach in such scenarios.
> > >
> > > Thank you again for your constructive suggestions and for increasing your score.
> > >
> > > Best regards,
> > >
> > > Authors

---

### Official Review · Reviewer_LZPM · 2024-07-11

**Soundness:** 3
**Presentation:** 3
**Contribution:** 3
**Rating:** 8
**Confidence:** 4

**Summary:**

This paper introduces MMSite, a multi-modal framework to improve the identification of active sites in proteins by leveraging protein sequences and textual descriptions. The authors build the ProTAD, a dataset containing over 570,000 pairs of protein sequences and detailed textual descriptions. The MMSite uses a “First Align, Then Fuse” strategy to align textual descriptions with protein sequences and fuses the modalities to enhance the identification of active sites. It also employs manual prompting, MACross module and soft-label alignment during the alignment phase. Based on this framework, MMSite outperforms existing protein representation learning methods on several metrics.

**Strengths:**

This paper is well-written and clearly structured, making it easy to read and understand. The paper proposes an important task in the field of biological science. Faced with the scarcity of per-residue labeled data, the authors innovatively employ detailed textual descriptions to assist evolutionary scale models in the identification of active sites. The carefully designed framework addresses the challenges of integrating multiple data modalities through a “First Align, Then Fuse” strategy. Experimental validation results show the state-of-the-art performance of MMSite compared to other baselines, and demonstrate the robustness of the framework. In addition, the authors build a large dataset covering a wide range of protein attributes, which is helpful for subsequent protein related research.

**Weaknesses:**

1. The framework relies on a complex multi-modal integration mechanism, and in the reference stage, an agent model is used to generate text modality, which may lead to high computational costs.

2. In the last raw of Tables 1, 8 and 9, MMSite also belongs to Seq. & Text model, but it is separated from that cell. In Table 4, the way of indicating different training strategies by different colors is not mainstream.

3. Some specific details in this paper are not clearly explained, as listed in the Questions part.

**Questions:**

1. In Tables 1, 8 and 9, for ProtST (w/ and w/o retrain), which is also a protein-text multimodal model, why does it perform so much worse than MMSite?

2. In Section 3.2, you “adopt cross-modal soft-label alignment to make one-hot label continuous”, but in the reference [19] you cite, the original authors also adopted another uni-modal soft-label alignment, which is also useful to improve the performance in the case of soft-label. Have you tried this approach?

3. The ProTAD dataset contains over 570,000 pairs of protein sequences and textual descriptions, but why does your training set have less than 50,000 samples as shown in Table 6?

**Limitations:**

The authors have addressed the limitations mentioned in the main text and appendix. To reduce the potential negative societal impact of the misuse of the model, the authors have restricted the license in the supplementary material.

---

> ### Author Rebuttal · Authors · 2024-08-07
>
> Dear Reviewer,
>
> Thank you for your valuable feedback and constructive suggestions. Below is our point-by-point response to your comments.
>
> **Re - Weaknesses 1**
>
> Thanks for your concern regarding the inference efficiency of our model! During inference, if the protein has corresponding multi-attribute text descriptions, the process is the same as during training. However, if such descriptions are not available, we use the Prot2Text agent model to generate the text inputs. Below is a comparison of MMSite inference time (seconds per sample) between using pre-existing text descriptions from ProTAD and generated text with an agent model. For further context, we also compare against the BioMedGPT [1], another excellent protein-to-text model. The tests were conducted on a single GPU (NVIDIA GeForce RTX 4090) and a CPU (Intel(R) Xeon(R) Platinum 8375C CPU @ 2.90GHz).
> |Source of Text|Average GPU time (s/sample)|Average CPU time (s/sample)|
> |:-:|:-:|:-:|
> |ProTAD|0.1336|1.7523|
> |Prot2Text generation|0.9044|4.2963|
> |BioMedGPT generation|5.1136|71.1690|
>
> Additionally, we also compared the model performance using text generated by Prot2Text and BioMedGPT:
> |Source of Text|F_max|AUPRC|MCC|OS|FPR|
> |:-:|:-:|:-:|:-:|:-:|:-:|
> |Prot2Text generation|**0.8250**|0.8909|**0.8319**|**0.8549**|**0.1689**|
> |BioMedGPT generation|0.8230|**0.8921**|0.8304|0.8540|0.1693|
>
> As the comparison shows, while using Prot2Text does increases the inference time of ProTAD, it provides comparable performance to BioMedGPT with a much smaller inference time cost. We will add this discussion to the revised paper.
>
> **Re - Weaknesses 2**
>
> Thank you for your careful review and constructive suggestions! You are correct that MMSite should be classified as a Seq. & Text model, and we will revise the tables accordingly. We will address these issues in future versions. Thank you again for your valuable feedback.
>
> **Re - Weaknesses 3**
>
> Please refer to the responses in Re - Questions 1-3 sections.
>
> **Re - Questions 1**
>
> Thank you for your insightful question! While ProtST is also a protein-text multimodal model, it aligns global sequence-level features with textual descriptions during training, rather than focusing on token-level features. This approach overlooks individual amino acid characteristics, causing deviations that reduce their effectiveness in reflecting inherent properties, ultimately resulting in poorer performance. In contrast, MMSite integrates text-empowered sequence representations directly at the token level. By using the MACross module and skip concatenation, we preserve the original features of amino acids while enriching them with relevant textual context. This approach ensures that amino acid representations maintain their intrinsic properties and benefit from the additional information provided by the text descriptions.
>
> **Re - Questions 2**
>
> Thank you for your careful review of our paper! In the reference you mentioned (Prot2Text), the original authors adopted both cross-modal and uni-modal soft-label alignment. The uni-modal alignment was proposed to address the issue where, despite alignment between two modalities in a cross-modal setting, the feature distribution within the same modality could deviate. This deviation may cause non-corresponding samples from the same modality to be very close in the feature space, leading to incorrect retrieval. However, in our task, retrieval is not a concern, and incorporating an additional uni-modal alignment loss could lead to suboptimal performance. To evaluate this, we conducted a comparison experiment:
> |Method|F_max|AUPRC|MCC|OS|FPR|
> |:-:|:-:|:-:|:-:|:-:|:-:|
> |w/o uni-modal soft-label alignment|**0.8250**|**0.8909**|**0.8319**|**0.8549**|**0.1689**|
> |w/ uni-modal soft-label alignment|0.8095|0.8739|0.8172|0.8416|0.1829|
>
> As shown in the table, adding uni-modal soft-label alignment resulted in decreased performance across all metrics. Therefore, we opted not to include this additional alignment in our final approach.
>
> **Re - Questions 3**
>
> Thank you for your insightful question! We built the ProTAD dataset based on UniProt, and it contains more than 570,000 pairs of protein sequences and textual descriptions. To ensure diversity and avoid redundancy, we used MMSeqs2 to cluster these sequences based on a sequence identity threshold. We then randomly selected one sequence from each cluster to construct the final dataset (as illustrated in Appendix A.2). Therefore, the size of the training set is determined by the number of clusters, which in turn depends on the clustering threshold.
>
> Thank you again for your time and consideration.
>
> Best regards,
>
> Authors
>
> **Reference**
>
> [1] Zhang K, Yu J, Yan Z, et al. BiomedGPT: A Unified and Generalist Biomedical Generative Pre-trained Transformer for Vision, Language, and Multimodal Tasks. arXiv, 2023.

---

> > ### Comment · Reviewer_LZPM · 2024-08-14
> > **All of my concerns have been addressed and I like this work**
> >
> > The authors have made a great effort during the rebuttal. All concerns have been addressed. Consequently, I raised the score and recommended acceptance of this submission.

---

> > > ### Author Response · Authors · 2024-08-14
> > >
> > > We sincerely appreciate your thoughtful consideration and positive recommendation. Thank you for your time and effort in reviewing our work.

---

### Official Review · Reviewer_zjg9 · 2024-07-11

**Soundness:** 2
**Presentation:** 3
**Contribution:** 3
**Rating:** 6
**Confidence:** 3

**Summary:**

The paper constructs a Protein-attribute text dataset, ProTAD, and proposes a multi-modal framework, MMsite, that enhances PLM with biomedical language models. During the inference stage, the authors propose generating biomedical text with Prot2Text, which is then fed into the MMsite module. The paper demonstrates that BLM enhances PLM’s performance on protein active sites identification, which is less studied in protein representation learning settings, through various experiments.

**Strengths:**

- The paper demonstrates that BLM enhances PLM’s performance through various experiments. It can be integrated with different PLMs and BLMs in future studies.

- The cross-modal soft-label alignment and multi-modal fusion for prediction is a novel contribution that could be adopted for the fusion of diverse models.

**Weaknesses:**

- Experiments are mainly performed on one dataset. More general protein-level tasks or residue-level tasks should be included, such as protein fitness prediction, protein localization prediction, protein function annotation, and binding site prediction.

**Questions:**

1. **Generalizability**: Will the MMsite model help with other general protein representation learning tasks? We hope to gain more insights into the incorporation of biomedical textual data.

2. I believe that the ProtDescribe [1] dataset has already built a protein-text dataset from Swiss-Prot. Could you explain the necessity and advantage of your work?

3. In the experiment table, ProtST [1] is not better than non-finetuned ESM? Normally, ProtST-induced PLMs outperform the vanilla PLMs. Could you provide an explanation?

4. On line 176, you mentioned, “However, in our case, there may be a potential semantic association between unpaired sequences and textual descriptions in the same batch due to the principle ‘Similar protein sequences give rise to similar structures and functions’ in biology.” Could you elaborate on how you curate the batches?

5. During inference, the Prot2Text model uses an ESM2 model and a GPT-2 decoder, which I suppose would be time-consuming. I hope the authors can discuss this issue.

[1] Xu M, Yuan X, Miret S, et al. Protst: Multi-modality learning of protein sequences and biomedical texts[C]//International Conference on Machine Learning. PMLR, 2023: 38749-38767.

**Limitations:**

Please refer to Weaknesses and Questions sections.

---

> ### Author Rebuttal · Authors · 2024-08-07
>
> Dear Reviewer,
>
> Thank you for your valuable feedback and constructive suggestions. Below is our point-by-point response to your comments.
>
> **Re - Weaknesses**
>
> To the best of our knowledge, ProTAD is currently the only dataset that simultaneously includes amino acid-level labels and detailed protein textual descriptions, making direct comparisons with other datasets challenging. Regarding the general protein-level tasks you mentioned, our MMSite framework incorporates comprehensive and rich textual descriptions of proteins as input. These descriptions inherently contain information about function and/or localization, making evaluations on these tasks somewhat unreasonable. In contrast, the annotation of active sites is not explicitly included in the input, which justifies the focus on active site prediction in our work. As for binding site prediction, it is not suitable to rely solely on protein sequence, as the binding sites often depend on specific interactions with various small molecules, which are not included in our model's input.
>
> **Re - Questions 1**
>
> Please refer to our response in the Re - Weaknesses section above.
>
> **Re - Questions 2**
>
> Thank you for mentioning an important related work! Here we compare our ProTAD dataset with ProtDescribe:
> Firstly, ProtDescribe includes 4 text attributes, whereas ProTAD comprises 17 text attributes, providing more comprehensive and detailed information.
> Secondly, ProTAD includes fine-grained amino acid-level labels, which are not present in ProtDescribe. These labels are crucial for the residue-level tasks such as active site identification from a multimodal perspective.
>
> **Re - Questions 3**
>
> ProtST-induced PLMs do indeed outperform vanilla PLMs in many downstream, sequence-level tasks, such as protein localization prediction and function annotation.
> This is because ProtST aligns the protein-level sequence embeddings with text during training, rather than token-level features, causing the representations to focus on the global function of the protein.
> We suspect that by doing so, the model loses sharpness in the individual characteristics of each amino acid, which reduces their effectiveness in residue-level tasks.
>
> **Re - Questions 4**
>
> In fact, we follow the standard practice of randomly sampling each batch from the training set. However, considering this biological principle, we use soft-label alignment to address the limitations of traditional hard-label alignment (such as CLIP [1]), where positive pairs are assigned a label of 1, and negative pairs are assigned a label of 0. The model then “pulls close” the positive pairs and “pushes away” the negative pairs in the high-dimensional space with equal emphasis to each pair. However, for similar protein sequences $S_a$ and $S_b$, their corresponding texts $T_a$ and $T_b$ may also be similar. Thus, pushing away $S_a$ and $T_b$ may not always be reasonable. To mitigate this issue, we employ soft-label alignment, which aligns the distribution $Q_i^{\rm s2t}$ towards $P_i^{\rm s2s}$ and $Q_i^{\rm t2s}$ towards $P_i^{\rm t2t}$ as described in Equation 8. This approach allows for a more nuanced alignment that respects the inherent similarities within the sequence and textual domains.
>
> **Re - Questions 5**
>
> Thank you for your valuable suggestion!
> During inference, if the protein has corresponding multi-attribute text descriptions, the process is the same as during training. However, if such descriptions are not available, we use the Prot2Text agent model to generate the text inputs. Below is a comparison of MMSite inference time (seconds per sample) between using pre-existing text descriptions from ProTAD and generated text with an agent model. For further context, we also compare against the BioMedGPT [2], another excellent protein-to-text model. The tests were conducted on a single GPU (NVIDIA GeForce RTX 4090) and a CPU (Intel(R) Xeon(R) Platinum 8375C CPU @ 2.90GHz).
> |Source of Text|Average GPU time (s/sample)|Average CPU time (s/sample)|
> |:-:|:-:|:-:|
> |ProTAD|0.1336|1.7523|
> |Prot2Text generation|0.9044|4.2963|
> |BioMedGPT generation|5.1136|71.1690|
>
> We also compared the model performance using text generated by Prot2Text and BioMedGPT:
> |Source of Text|F_max|AUPRC|MCC|OS|FPR|
> |:-:|:-:|:-:|:-:|:-:|:-:|
> |Prot2Text generation|**0.8250**|0.8909|**0.8319**|**0.8549**|**0.1689**|
> |BioMedGPT generation|0.8230|**0.8921**|0.8304|0.8540|0.1693|
>
> As the comparison shows, while using Prot2Text does increases the inference time of ProTAD, it provides comparable performance to BioMedGPT with a much smaller inference time cost. We will add this discussion to the revised paper.
>
> Thank you again for your time and consideration.
>
> Best regards,
>
> Authors
>
> **Reference**
>
> [1] Radford A, Kim J W, Hallacy C, et al. Learning Transferable Visual Models From Natural Language Supervision. ICML, 2021.
>
> [2] Zhang K, Yu J, Yan Z, et al. BiomedGPT: A Unified and Generalist Biomedical Generative Pre-trained Transformer for Vision, Language, and Multimodal Tasks. arXiv, 2023.

---

> > ### Comment · Reviewer_zjg9 · 2024-08-13
> >
> > Thank you for your detailed responses. I believe some of my concerns have been addressed, and I have raised my score to 6.
> > However, I still feel that the work's originality compared to previous research is marginal.

---

> > > ### Author Response · Authors · 2024-08-14
> > >
> > > We are pleased to hear that the rebuttal has addressed your concerns and appreciate that you have raised your score!
> > > While our approach of using text descriptions to enhance protein representations may seem similar to previous research, our innovation lies in the integration of protein sequences and biomedical texts for fine-grained, **token-level** prediction of active sites. To the best of our knowledge, this is the first work to tackle this task in such a manner. Given the abundance of text available in the biomedical domain, we believe that our work has the potential to inspire further impactful research in this direction and lead to breakthroughs in important but data-scarce tasks.

---

### Official Review · Reviewer_CdPy · 2024-07-12

**Soundness:** 2
**Presentation:** 2
**Contribution:** 2
**Rating:** 4
**Confidence:** 4

**Summary:**

The paper proposes a framework to improve the active site prediction for the protein representations by fine-tuning the model on the text function descriptions.

**Strengths:**

* The paper addresses active site prediction, which is an interesting and understudied problem in protein science.
* The study compares its proposed method with numerous baselines.

**Weaknesses:**

* Limited Method Novelty: The paper's core idea of using text descriptions to enhance protein representations lacks originality. Similar approaches have been previously explored in works such as ProtST (Xu 2023) and ProteinCLAP (Liu 2023). Given that this is the major component of the paper, the method appears incremental.
* Insufficient Motivation: The paper fails to provide a clear rationale for training on 17 different types of function annotations to improve active site predictions. Many of these annotations, such as organism information, seem irrelevant to the task at hand. Furthermore, the paper does not adequately explain how protein-level annotations are expected to benefit amino acid-level predictions.
* Lack of Temporal Evaluation: While the paper adopts a structural split for evaluation, which is acceptable, a temporal-based evaluation would be more ideal and realistic. A temporal split, where some proteins are held out based on their discovery time, would more accurately reflect real-world scenarios in scientific applications.
* Poor Writing Quality: The overall writing quality of the paper is subpar, particularly in the methods section. The explanation of methodologies is convoluted and difficult to follow.

**Questions:**

ProtST is also trained on text descriptions. What do you think makes your method outperforms ProtST?

---

> ### Author Rebuttal · Authors · 2024-08-07
>
> Dear Reviewer,
>
> Thank you for your valuable feedback and constructive suggestions. Below is our point-by-point response to your comments.
>
> **Re - Weaknesses 1**
>
> We agree that using text descriptions to enhance protein representations may seem similar to existing works. However, our innovation lies in the integration of protein sequences and biomedical texts for the **token-level** prediction task of active site identification. Experiments show that our method outperforms existing multi-modal methods for sequence-level prediction tasks. As there are abundant text available in the biomedical domain, we anticipate our work will inspire more impactful research in this direction and lead to breakthroughs in important but data-scarce tasks.
>
> **Re - Weaknesses 2**
>
> Regarding the motivation of MACross, it is based on our judgement that the Function attribute is the most relevant attribute for predicting active sites and contains the richest information. Therefore, in our MACross module, we prioritize the Function attribute and integrate the remaining annotations through cross-attention mechanisms. This ensures the rich information from the Function attribute is complemented by other annotations.
>
> The additional attributes, although seemingly insignificant, also contribute to improve active site predictions. For example, subcellular location provides context about protein accessibility and potential interactions, critical for identifying active sites. Similarly, organism information offers evolutionary context, highlighting conserved or mutated motifs. The table below shows the performance comparison between using only the Function attribute and using all attributes, demonstrating that incorporating all attributes leads to better performance.
> |Textual Description|F_max|AUPRC|MCC|OS|FPR|
> |:-:|:-:|:-:|:-:|:-:|:-:|
> |All attributes|**0.8250**|**0.8909**|**0.8319**|**0.8549**|**0.1689**|
> |Only Function attribute|0.8152|0.8866|0.8231|0.8471|0.1764|
>
> Regarding how protein-level annotations help, we apologize for not having explained this clearly in the paper. In the rich study (i.e. abundant text) of the function of proteins, scientists have paied special attention to the identification, structural analysis, engineering of active sites. These valuable information are stored not in protein sequences but in biomedical texts. We expect BLMs to retrieve this information given protein-level annotations, providing context for active site prediction.
>
> **Re - Weaknesses 3**
>
> We appreciate your suggestion for a temporal-based evaluation. To address this, we conducted an additional experiment simulating the discovery of new proteins. Our ProTAD dataset includes data up to March 11, 2024. We collected data from UniProt recorded after this date, representing newly discovered proteins (115 samples with active site labels), to evaluate the model's performance. The results show that MMSite maintains high performance even on newly discovered proteins, as shown in the table below.
> |Test Data|F_max|AUPRC|MCC|OS|FPR|
> |:-:|:-:|:-:|:-:|:-:|:-:|
> |Newly discovered proteins|0.8432|0.8865|0.8460|0.8465|0.1420|
>
> **Re - Weaknesses 4**
>
> We apologize for the shortcomings in our writing. Here we provide a clearer explanation of our methodologies in the Methods Section:
> - In Section 3.1, we formulate our task of predicting active sites in proteins using both protein sequences and multi-attribute textual descriptions.
> - In Section 3.2, we detail our methodology through four main steps:
>   - **Attribute description reconstruction with prompt:** We use manual prompting to reconstruct multi-attribute text descriptions into a format processable by BLMs.
>   - **Modality feature extraction:** We extract features from both protein sequences and textual descriptions using a PLM and a BLM, respectively. In the MACross module, we separately process the Function attribute and integrate it with the remaining attributes using cross-attention.
>   - **Cross-modal soft-label alignment:** Our two-stage training strategy, "First Align, Then Fuse", uses soft-label alignment to align features from two modalities. This method differs from the traditional contrastive approaches that "pull close" positive pairs and "push away" negative pairs, which may not be suitable when there is significant similarity between different sequences or texts in a batch.
>   - **Multi-modal fusion and active site identification:** We then fuse the two modalities through fusion attention and skip concatenation to predict active sites. In addition, since newly discovered proteins may lack corresponding text descriptions during inference, we use an agent model to generate the necessary texts, which are then used alongside the sequences.
>
> Thank you for the opportunity to present our writing structure, and we will reorganize the content of this section to enhance the clarity and readability of our work in future versions.
>
> **Re - Questions**
>
> While ProtST also uses text descriptions, it aligns global protein-level sequence representations with textual descriptions during its training stage, causing the representations to focus on the global function of the protein. This training approach overlooks individual amino acid features, which reduces their effectiveness in residue-level tasks. In contrast, MMSite integrates text-empowered sequence representations directly at the token level. By using the MACross module and skip concatenation, we preserve the original features of amino acids while enriching them with relevant textual context. This approach ensures that amino acid representations maintain their intrinsic properties and benefits from the added information provided by the text descriptions.
>
> Thank you again for your time and consideration.
>
> Best regards,
>
> Authors

---

> ### Comment · Reviewer_CdPy · 2024-08-13
>
> Thank you for addressing some of my concerns. I've updated my score to reflect these clarifications. However, I still have two main concerns:
>
> Novelty: The work's originality compared to previous research still feels marginal.
>
> Dataset motivation: The rationale for collecting such a large dataset with 17 attributes is not fully justified, especially given the experimental results. The new experiment shows that training on the function attribute alone achieves similar performance to using all 17 attributes. This raises questions about the necessity and efficiency of the full dataset.

---

> > ### Author Response · Authors · 2024-08-14
> >
> > We are glad to hear that the rebuttal addresses your concerns, and also glad that you have raised your score. Regarding the two points you mentioned:
> >
> > **Novelty:** In this work, we integrate protein sequences with biomedical text to achieve fine-grained, **token-level** prediction of active sites, which, to the best of our knowledge, is the first to tackle this task in such a manner. We believe that our approach of using text descriptions to enhance protein representations for residue-level learning has the potential to inspire further impactful research in the biomedical domain.
> >
> > **Dataset motivation:** While the new experiment shows similar performance, training with all attributes still yields better results across all metrics. Our rationale for collecting such a large dataset was to capture a broader range of contextual information, which can be beneficial in more complex scenarios or when specific attributes provide unique insights not captured by the Function attribute alone. In addition, a rich dataset with diverse attributes provides greater flexibility for future experiments and allows for fine-tuning to explore how these attributes interact and contribute to performance in different contexts.
> >
> > We will continue to refine our work in the future. Thank you again for your valuable comments!

---

### Official Review · Reviewer_9yAm · 2024-07-12

**Soundness:** 3
**Presentation:** 3
**Contribution:** 3
**Rating:** 5
**Confidence:** 3

**Summary:**

This paper studies the problem of identifying active sites in proteins.
The paper proposes a novel multi-modal framework, MMSite that aligns and fuses textual and sequential modalities of protein sequences ("First Align, Then Fuse" strategy). The authors leverage pre-trained protein language models and biomedical language models for extracting features from protein sequences and attribute textual descriptions.
For this training, they introduce their newly constructed ProTein-Attribute text Dataset (ProTAD) and a MACross attention module for combining multiple attributes.
Via empirical evaluation, they show that MMSite can significantly enhance the identification of active sites in proteins compared to existing protein representation learning methods, e.g., ESM, and ProtST. The paper also provides a set of ablation studies for a better understanding of the effect of frameworks' components and configuration.

**Strengths:**

1. Their construction of a large-scale, multi-modal dataset for ProTein-Attribute association is valuable for further research in this field.
2. The multi-modal framework MMSite is somewhat novel in the context of active site prediction. This idea of multi-modal has been explored recently for protein studies, e.g., ProteinCLIP (https://www.biorxiv.org/content/10.1101/2024.05.14.594226v1)
3. The paper demonstrates significant performance improvement on the evaluation testbed.
4. This paper provides a good and thorough set of ablation studies and examples (in the Appendix) that help understand the framework and its working scenarios.

**Weaknesses:**

My main concern is about the **evaluation**:
1. Baselines: the paper only compares the framework PRL adapted with four trainable Transformer layers which are smaller in size compared to the MMSite and not designed for the task. Furthermore, there are some PLMs with better performance than the baselines, e.g., SaProt (https://www.biorxiv.org/content/10.1101/2023.10.01.560349v5.full.pdf)
How about the existing active sites prediction methods without the need for text mentioned in the paper? e.g., classical methods (Random Forest and Support Vector Machine), statistics-based approaches, DeepSurf, ... or methods using structure instead of text?
2. Dataset: The dataset (ProTAD) might have inherent biases that are not discussed in the paper. If the dataset is not diverse enough, the model may perform well on the dataset but fail to generalize to other proteins. If the paper can provide another dataset, that would me more convincing.
3. In table 9, for dataset with clustering threshold at 50%, we observe the base PLM models are close to MMSite and can outperform MMSite in some metrics. So, is that the case MMSite heavily depends on the base model and works well only with low clustering thresholds?

**Methods:**
* The novelty is somewhat limited in terms of machine learning methodology as the paper heavily relies on existing methods like PLMs, BLMs, and cross-attention mechanisms. The use of soft-label alignment is applied in a new context.
From my understanding, besides the new dataset, the main novelty is the design of the MMSite framework for the biology application.
* The model works with short sequence (512 amino acids)

**Claims:**
* The claim that their method "enhances the identification capabilities of PLMs" should be elaborated more clearly to the context as the MMSite, to my understanding, does not improve the model, rather than just use it.
* Some biology claims are without reference and need further explanation, especially for readers with primary machine learning background. For examples, (lines 177--179) "... due to the principle “Similar protein sequences give rise to similar structures and functions” in biology."

**Minors:**
* Some figures may need to be improved, both in quality and presentation. For example, in figure 1, the dash lines are not well aligned  and the introduction of "Mark Zuckerberg" figure seems not necessary.
* "soft-label" and "soft label" inconsistency format
* In figure 2, $f_{\Phi}$ is denoted PLM, but seem like $f_{\phi}$ is used in the main text.
* Typos:
    * Line 105, "Sites" -> "sites"
    * Fig1 caption: "mutual tasks", is that "multi-modal"?
    * abstract: "life science" -> the life sciences

**Questions:**

Together with the above questions, I have the following questions,

1. What is the accuracy/performance of the Pro2Text on the dataset? We can test this on the proteins that the text exists.
2. How does the MMSite perform when we use Pro2Text to generate texts for training instead of the textual description from the dataset?
3. For text processing, how do you process the Nan attributes? Are you using Pro2Text to fill in?
4. For the alignment phase, what happens if some proteins in the batch are 10% different while their texts are similar?
5. In Figure 1, the MACross is skipped when texts are not available and in Table 3, the gain of MACRoss seems marginal. Have you evaluated the model without MACross and used the prompt to connect multiple attributes instead?

**Limitations:**

No.
The paper lacks a discussion on the limitations and impact, especially for their practical applications, e.g., discussions regarding the potential misuse of the technology in drug design and bioengineering.
The paper should discuss the scenarios when MMSite works well, e.g., dependency on the base PLM, agent models, and clustering threshold.

---

> ### Author Rebuttal · Authors · 2024-08-07
>
> Dear Reviewer,
>
> Thank you for your valuable feedback and constructive suggestions. Below is our point-by-point response to your comments.
>
> **Re - Weaknesses Evaluation 1**
>
> Regarding # of trainable params, the difference between MMSite and baselines is not very large, as the former has 9 trainable Transformer layers for sequence and the latter have 4. We are running experiments with 9 layers for baselines for fair comparison and will report results once they finish.
>
> Regarding comparison with other baselines, we conducted experiments with SaProt, Random Forest (RF), Support Vector Machine (SVM), and DISCERN (a stat-based method designed for active site prediction). The results are shown below, with MMSite significantly outperforming them.
> ||F_max|AUPRC|MCC|OS|FPR|
> |:-:|:-:|:-:|:-:|:-:|:-:|
> |MMSite|**0.8250**|**0.8909**|**0.8319**|**0.8549**|**0.1689**|
> |SaProt (650M-AF2)|0.7181|0.8562|0.7259|0.7480|0.2731|
> |RF|0.4339|0.5545|0.4382|0.3137|0.2825|
> |SVM|0.4270|0.4708|0.4017|0.3409|0.4000|
> |DISCERN|0.2870|0.4539|0.2949|0.1921|0.4176|
>
> For DeepSurf, it is specifically designed for predicting binding sites with molecules, which is not applicable to our task. Besides, we have already compared methods that use structure (e.g. MIF and PST).
>
> **Re - Weaknesses Evaluation 2**
>
> We believe ProTAD is diverse enough, as it is derived from Swiss-Prot, referred to as "a reference resource of protein sequences and functional annotation that covers proteins from all branches of the tree of life" [1]. All Swiss-Prot data with both active site and function annotations are included without cherry-picking. The split strategy is strict (10% sequence similarity), which challenges the generalization ability.
>
> **Re - Weaknesses Evaluation 3**
>
> MMSite depends on the base model, but when the split strategy is harsh (e.g. 10% sequence similarity), the base models couldn't generalize, whereas MMSite can find meaningful patterns from both sequence and text to achieve good performance. At 50% threshold, their performances are close because (1) the dataset is easy enough for base models to generalize, and (2) the performance is already very high, leaving little room to improve.
>
> **Re - Weaknesses Methods 1**
>
> We agree that the novelty of MMSite mainly lies in the design of the framework and its application. However, we believe this does not undermine the significance of our work, which is among the first to integrate protein sequences and biomedical texts for fine-grained, residue-level prediction. As there is abundant text available in the biomedical domain, we anticipate our work will inspire more impactful research and lead to breakthroughs in important but data-scarce tasks.
>
> **Re - Weaknesses Methods 2**
>
> In Appendix C.1, we evaluated MMSite across multiple sequence lengths (from 128 to 1024), and the results show its robust and consistent performance.
>
> **Re - Weaknesses Claims**
>
> Thanks for your suggestions. We will revise our claim to clearly reflect how MMSite builds, and provide more context to help readers better understand the underlying biological rationale in the future version.
>
> **Re - Weaknesses Minors**
>
> We have updated Figure 1 in the submitted PDF, and we will correct the words and notations in the future version. Besides, "mutual tasks" refers to the tasks of generating texts from sequence and retrieving sequences from text. We have revised it in the PDF.
>
> **Re - Questions 1**
>
> To evaluate the performance of Prot2Text, we built a dataset of 1k proteins randomly sampled from ProTAD, as shown below. The performance on ProTAD-1k is good and generally consistent with the reported performance, indicating that Prot2Text can readily serve as an agent to generate textual descriptions for proteins.
> ||BLEU Score|Rouge-1|Rouge-2|Rouge-L|BERT Score|
> |:-:|:-:|:-:|:-:|:-:|:-:|
> |Prot2Text (reported)|**36.51**|53.68|45.60|51.40|85.20|
> |Prot2Text on ProTAD-1k|25.73|**56.56**|**49.46**|**54.15**|**86.57**|
>
> **Re - Questions 2**
>
> The table below compares the performance of MMSite using human-annotated and Prot2Text-generated texts for training. The performance of two settings was similar across most metrics.
> |Source of Text|F_max|AUPRC|MCC|OS|FPR|
> |:-:|:-:|:-:|:-:|:-:|:-:|
> |ProTAD (human annotation)|**0.8250**|0.8909|**0.8319**|**0.8549**|**0.1689**|
> |Prot2Text generation|0.8194|**0.8923**|0.8254|0.8433|0.1751|
>
> **Re - Questions 3**
>
> For missing values, we use "unknown" as the textual annotation. (Appendix A.1)
>
> **Re - Questions 4**
>
> Suppose proteins $i$ and $j$ have similar texts but different sequences. The core of the alignment loss is two KL terms, which pushes the distributions $Q_i^{\rm s2t}$ close to $P_i^{\rm s2s}$ and $Q_i^{\rm t2s}$ close to $P_i^{\rm t2t}$. This creates both attractive and repulsive forces between the sequence embedding for $i$ and the text embeddings for $j$, striking a balance between full alignment and full separation. Our soft-label alignment strategy ensures that the model can effectively learn the relationship between sequences and texts, even when the two modalities sometimes give conflicting signals.
>
> **Re - Questions 5**
>
> The ablation study in Table 3 have evaluated this scenario. It shows that adding MACross improves the model's performance across five metrics by about 1.7% on average. The main purpose of MACross is to highlight the importance of the Function attribute, because it contains richer information than others, which is essential for our task. Otherwise, the model may struggle to prioritize the important attributes, leading to suboptimal performance.
>
> **Re - Limitations**
>
> We discussed the limitations and impacts in Appendix D, and restricted the license to mitigate potential misuse in the code. We will discuss more scenarios in the future version.
>
> Thank you again for your time and consideration.
>
> Best regards,
>
> Authors
>
> **Reference**
>
> [1] Coudert E, Gehant S, De Castro E, et al. Annotation of biologically relevant ligands in UniProtKB using ChEBI. Bioinformatics, 2023.

---

> > ### Comment · Reviewer_9yAm · 2024-08-07
> > **Re: Rebuttal by Authors**
> >
> > Thank you for the extra results and detailed responses.
> >
> > Given the similar performance between Prot2Text-generated texts and human-annotated texts, should we directly use Prot2Text-generated texts which is much more cost-efficient? or do we have any scenarios for which human-annotated texts are preferable?

---

> > > ### Author Response · Authors · 2024-08-08
> > > **Response and More Evaluation Results**
> > >
> > > **Response to Your Question**
> > >
> > > Thank you for your question. The table in **Re - Questions 2** compares the performance when trained with human-annotated texts versus Prot2Text-generated texts. While the performance is similar, training with human-annotated texts yields better results across most metrics, as they provide more accurate and comprehensive information. Furthermore, the comparable performance also indicates that Prot2Text is capable of effectively generating textual descriptions, particularly for newly discovered proteins that might lack annotations.
> > >
> > > Therefore, we recommend using human-annotated data during training to fully integrate the detailed information provided by these texts. For inference, if human-annotated texts are available, they should be used to better align with the model’s pre-training and leverage the pre-trained knowledge more effectively. Otherwise, Prot2Text-generated texts can serve as a suitable alternative.
> > >
> > > **More Evaluation Results for *Re - Weakness Evaluation 1***
> > >
> > > In addition, as a supplement to our response for **Weakness Evaluation 1**, we adjusted the number of trainable Transformer layers followed by the baseline models to ensure that the number of trainable parameters is comparable to those in MMSite. The results are shown in the table below:
> > >
> > > |       Method & Version       |   F_max    |   AUPRC    |    MCC     |     OS     |    FPR     |
> > > | :--------------------------: | :--------: | :--------: | :--------: | :--------: | :--------: |
> > > |            ESM-1b            |   0.7119   |   0.7420   |   0.7207   |   0.7492   |   0.2673   |
> > > |            ESM-1v            |   0.5706   |   0.8776   |   0.5775   |   0.5846   |   0.4053   |
> > > |          ESM-2-650M          |   0.6936   |   0.7696   |   0.6014   |   0.6211   |   0.3886   |
> > > |          ProtT5-BFD          |   0.4211   |   0.6727   |   0.4353   |   0.3705   |   0.5702   |
> > > |        ProtT5-UniRef         |   0.4262   |   0.6762   |   0.5382   |   0.5437   |   0.5652   |
> > > |         ProtBert-BFD         |   0.5603   |   0.7073   |   0.5663   |   0.5557   |   0.4063   |
> > > |       ProtBert-UniRef        |   0.4638   |   0.7204   |   0.4885   |   0.5227   |   0.4400   |
> > > |          ProtAlbert          |   0.6698   |   0.7284   |   0.6727   |   0.6833   |   0.3246   |
> > > |          ProtXLNet           |   0.0126   |   0.1270   |   0.0364   |   0.0535   |   0.9401   |
> > > |         ProtElectra          |   0.5869   |   0.6579   |   0.5976   |   0.5891   |   0.5633   |
> > > |        PETA-deep_base        |   0.5458   |   0.7941   |   0.5504   |   0.5498   |   0.4337   |
> > > |            S-PLM             |   0.6745   |   0.8443   |   0.6817   |   0.6811   |   0.2950   |
> > > |             TAPE             |   0.3270   |   0.5978   |   0.3293   |   0.3239   |   0.6598   |
> > > |             MIF              |   0.1184   |   0.2306   |   0.1316   |   0.2739   |   0.8784   |
> > > |            MIF-ST            |   0.1218   |   0.2337   |   0.1372   |   0.2646   |   0.8696   |
> > > |           PST-t33            |   0.6514   |   0.8312   |   0.6621   |   0.7007   |   0.3271   |
> > > |          PST-t33_so          |   0.6598   |   0.8163   |   0.6697   |   0.6990   |   0.3195   |
> > > | ProtST (w/ retrain) -ESM-1b  |   0.4201   |   0.6518   |   0.4353   |   0.3880   |   0.5829   |
> > > |  ProtST (w/ retrain) -ESM-2  |   0.1340   |   0.4660   |   0.1349   |   0.1370   |   0.7688   |
> > > | ProtST (w/o retrain) -ESM-1b |   0.4770   |   0.7321   |   0.4915   |   0.5122   |   0.3419   |
> > > | ProtST (w/o retrain) -ESM-2  |   0.3375   |   0.6982   |   0.5462   |   0.5313   |   0.4136   |
> > > |          **MMSite**          | **0.8250** | **0.8909** | **0.8319** | **0.8549** | **0.1689** |
> > >
> > > This comparison shows that even with a comparable number of trainable parameters, MMSite significantly outperforms other baseline models.
> > >
> > > Thank you again for your question and consideration.

---

> > > ### Author Response · Authors · 2024-08-14
> > >
> > > We would like to know if you have any additional concerns or suggestions regarding our work. If possible, we hope to engage in further discussion to earn your endorsement. If you are satisfied with our rebuttal and responses to your **Official Comment**, we kindly ask if you would consider raising your score. Thank you very much for your valuable feedback again.

---

### Author Rebuttal · Authors · 2024-08-07

Dear Reviewers,

We wish to extend our sincere gratitude for your time and insightful feedback on our manuscript. Your insights are greatly helpful in improving the quality and clarity of our work. The following is a summarized response to some of the valuable suggestions and common issues raised.

**1. Evaluation Suggestions**

Many reviewers' considerations regarding performance evaluation are valuable, including:

- Reviewer **9yAm**'s concern on the comparison with traditional methods, and the quality of texts generated by Prot2Text.
- Reviewer **CdPy**'s concern on the significance of different text attributes and performance on newly discovered proteins.
- Reviewers **zjg9** and **LZPM**'s concern on the model’s computational costs during inference, and the construction of loss function.
- Reviewer **sxh3**'s concern on the necessity of manual prompting.

We will include these validation experiments in the revised version of the paper to address these valuable insights.

**2. Comparison with ProtST**

- Reviewers **CdPy**, **zjg9**, **LZPM**, and **sxh3** raised questions about ProtST's performance compared with MMSite. ProtST aligns global protein-level sequence representations with text, missing individual amino acid details. MMSite, however, pay more attention to integrate text-empowered sequence representations with amino acid features, which preserves the intrinsic properties of amino acids and benefits from the additional information provided by text descriptions.

**3. Performance at Different Clustering Thresholds**

- Reviewers **9yAm** and **sxh3** noted that MMSite's performance is similar to baseline models at high clustering thresholds. This occurs because a higher clustering threshold results in more similarity between the training set and the test set. In a harsh split strategy, MMSite can identify meaningful patterns from both sequence and text, whereas baseline models cannot. Additionally, when the threshold is high, sequence-only models perform very well already, leaving little room for MMSite to improve further.

**4. Figures and Additional Clarifications:**

- Reviewers **9yAm** and **sxh3** suggested improvements for Figures 1 and 2. We have updated these figures in the uploaded PDF to more clearly reflect our work's aspects and MMSite's workflow.
- Additionally, we provide detailed dimension changes for each module to enhance understanding:
  - *Stage 1: Feature Extraction*
    - **PLM** extracts sequence features: $N^{\rm s}\times d^{\rm s}$ (sequence length × sequence dimension).
    - **BLM** extracts text features: $\\{N_i^{\rm t}\times d^{\rm t}\\}_{i=1}^{M}$ (length of $i$th attribute text × text dimension, for $M$ attributes).
    - A **Linear** layer maps the sequence features to $N^{\rm s}\times d^{\rm t}$.
    - In **MACross**, the [CLS] tokens of $M-1$ attributes (excluding Function) are concatenated to form a vector of dimension $(M-1)\times d^{\rm t}$ after **Inter-attribute Fusion**. Overall, MACross outputs a vector of dimension $(M-1)\times d^{\rm t}$.
  - *Stage 2: Semantic Alignment*
    - The **Shared Transformer Encoder** outputs the sequence and text features of dimension  $N^{\rm s}\times d^{\rm s}$ and $(M-1)\times d^{\rm t}$, respectively.
    - The first dimension is then averaged along for both modalities to obtain two vectors of $1\times d^{\rm t}$ for similarity calculation.
  - *Stage 3: Fusion and Prediction*
    - The **Fusion Attention** outputs the text-enhanced sequence representation with dimension $N^{\rm s}\times d^{\rm t}$.
    - The original sequence features are concatenated along the second dimension with the text-enhanced sequence, resulting in $N^{\rm s}\times (d^{\rm s}+d^{\rm t})$.
    - The **Active Site Prediction Head** maps this to $N^{\rm s}\times 2$ using an MLP.

Finally, we appreciate your detailed suggestions on acronyms, writing quality, notation, and figures. We will carefully follow your suggestions in the revised version to ensure they are correct and clear.

Thank you again for your valuable feedback.

Best regards,

Authors

---

### Decision · Program_Chairs · 2024-09-25

**Decision:**

Accept (poster)

**Comment:**

Reviewers are mildly positive about this paper; the fundamental criticism is that the research is rather incremental compared to previous work in this very hot research area. As the area chair, after reading the paper myself, I recommend acceptance. The authors have clearly done a lot of work and responded very carefully to the reviewers. The research should be made available to the community and the authors should receive credit for their hard work.

The scores from reviewers are 5, 4, 6, 8, 5. The authors criticize the review with score 4, but this review is sensible and its author is qualified. In contrast, the review with score 8 is rather superficial and its author is from a different research area. Therefore, as the area chair, I think that a reasonable consensus score is somewhere between 5 (the average without 8) and 5.6 (with).